# Learning from Online Videos at Inference Time for Computer-Use Agents

**Yujian Liu**[1]    **Ze Wang**[2]    **Hao Chen**[2]    **Ximeng Sun**[2]    **Xiaodong Yu**[2]
**Jialian Wu**[2]    **Jiang Liu**[2]    **Emad Barsoum**[2]    **Zicheng Liu**[2]    **Shiyu Chang**[1]

[1]**UC Santa Barbara**    [2]**Advanced Micro Devices, Inc.**
{yujianliu,chang87}@ucsb.edu

Reviewed on OpenReview: https://openreview.net/forum?id=YDFQIe6dqI

## Abstract

Computer-use agents can operate computers and automate laborious tasks, but despite recent rapid progress, they still lag behind human users, especially when tasks require domain-specific procedural knowledge about particular applications, platforms, and multi-step workflows. Humans can bridge this gap by watching video tutorials: we search, skim, and selectively imitate short segments that match our current subgoal. In this paper, we study how to enable computer-use agents to learn from online videos at inference time effectively. We propose a framework that retrieves and filters tutorial videos, converts them into structured demonstration trajectories, and dynamically selects trajectories as in-context guidance during execution. Particularly, using a VLM, we infer UI actions, segment videos into short subsequences of actions, and assign each subsequence a textual objective. At inference time, a two-stage selection mechanism dynamically chooses a single trajectory to add in context at each step, focusing the agent on the most helpful local guidance for its next decision. Experiments on two widely used benchmarks show that our framework consistently outperforms strong base agents and variants that use only textual tutorials or transcripts. Analyses highlight the importance of trajectory segmentation and selection, action filtering, and visual information, suggesting that abundant online videos can be systematically distilled into actionable guidance that improves computer-use agents at inference time. Our code is available at https://github.com/UCSB-NLP-Chang/video_demo.

## 1 Introduction

Computer-use agents promise to automate labor-intensive digital workflows, reshape how people interact with computers, and boost productivity across everyday and expert tasks (Deng et al., 2023; Xie et al., 2024; Zheng et al., 2024; PBC, 2024; OpenAI, 2025). Recent works have made rapid progress on a wide range of desktop and web tasks, yet a substantial gap remains between their performance and that of human users (Wang et al., 2025a; Agashe et al., 2025b; Wang et al., 2025b). We argue that a key reason for this gap is the agent's limited access to domain-specific procedural knowledge: how to operate particular applications, cope with idiosyncratic UI conventions, and execute multi-step workflows that depend on software version and platform. While agents continue to improve, progress on these capabilities is constrained by the scarcity of such data in large language models' (LLMs) training corpus.

On the contrary, humans are usually not limited by the lack of such domain knowledge. When we encounter unfamiliar applications or platforms, we rarely rely on our general knowledge alone. Instead, we consult online video tutorials and quickly learn useful information from the demonstrations. More specifically, we are able to search, skim, and selectively imitate video segments that match our immediate subgoals. These videos are abundant and richly visual, showcasing concrete UI manipulations that text-only resources frequently

omit. This observation motivates our central research question: *How can we enable computer-use agents to learn effectively from online videos at inference time?*

Answering this question is challenging for two reasons. First, there is a gap between watching videos and completing tasks as computer-use agents. Videos are continuous streams of frames, whereas agents issue actions as individual events, and they only observe sparse screenshots after completion of each action. Moreover, agents need to generate explicit UI actions, but actions performed in the videos are implicit and need to be inferred from screen transitions. Bridging from pixel changes in a video to actionable steps is therefore nontrivial. Second, raw videos are long and often contain digressions. During execution, an agent usually needs only a short, locally relevant snippet that matches its current observation and subgoal, so injecting entire videos can distract more than help.

To address these challenges, we introduce an inference-time learning-from-video framework that turns raw video tutorials into compact, actionable demonstrations the agent can consult step by step. Our approach has three components. First, given the task description, the agent generates search queries, searches online videos, and filters them to a small set of relevant tutorials. Second, a processing pipeline converts each filtered video into structured demonstration trajectories. Specifically, an off-the-shelf vision language model (VLM) infers underlying UI actions and synthesizes a concise objective for each subsequence of actions. With additional filtering of low-quality subsequences, this step yields a set of demonstration trajectories, each with a textual objective and a sequence of actions and observations. Third, a dynamic selection method ensures the agent consumes only what matters at the moment of decision. Particularly, before issuing the next action, the agent performs a coarse ranking of demonstrations followed by a detailed inspection of action lists to select a single trajectory that is most helpful for the next action. The selected trajectory is then provided in context to guide the agent's next decision. This design operationalizes how humans learn from videos: we search, skim, and imitate only the few seconds that matter. By converting videos into short, validated, visually grounded trajectories, the agent receives exactly the local guidance it needs, in the format it can act on.

We evaluate our method on OSWORLD-VERIFIED (Xie et al., 2024) and WEBARENA (Zhou et al., 2024), two widely used benchmarks for desktop and web tasks, respectively. Across both settings, our approach consistently outperforms strong agent baselines that lack video access, as well as variants that use only textual tutorials or video transcripts. For example, on OSWORLD-VERIFIED, our method outperforms the state-of-the-art `Jedi` framework (Xie et al., 2025b) by 2.1. On WEBARENA, our method outperforms `AgentOccam` (Yang et al., 2025b) by 4.2. Additional analyses show that our method benefits from more relevant videos, indicating the potential for further performance scale-up. Ablation studies also demonstrate the effectiveness of our designs, such as segmenting videos into short trajectories and dynamically selecting them, as well as filtering out irrelevant actions during video processing. Together, these results suggest that abundant online video tutorials can be systematically and automatically harvested and distilled into compact, actionable guidance that improves the ability of current computer-use agents.

## 2 Related Works

**Computer-use agents.** Recent advances in computer-use agents aim to endow LLM- or VLM-based systems with the ability to interact with GUIs on desktops and the web to complete tasks. Early work has focused on constructing realistic benchmarks and environments to support systematic evaluation (Xie et al., 2024; Deng et al., 2023; Zhou et al., 2024; Lin et al., 2024; Rawles et al., 2025; Bonatti et al., 2024; Zhao et al., 2025; Chen et al., 2025). Building on these benchmarks, subsequent systems explore different ways to couple multimodal perception with planning and control, including vision-enabled generalist agents, compositional generalist-specialist architectures, and reinforcement learning pipelines that improve agents' ability through training (Agashe et al., 2025b; Wang et al., 2025a; Qin et al., 2025; Wu et al., 2024; Yang et al., 2025c; Sun et al., 2025; Tan et al., 2025; Xu et al., 2025b; Xie et al., 2025a; Jia et al., 2025; Gao et al., 2024).

Similar to our work, some recent papers have investigated collecting demonstration trajectories and applying them in context or leveraging them for training. For example, Su et al. (2025) collects and filters trajectories from the agent's self-exploration in the environment. Ou et al. (2024) extracts trajectories from online tutorials, and Xu et al. (2025a) further simulates execution traces by following retrieved tutorials. However, self-exploration is limited by the agent's own capability, making it difficult to synthesize complex trajectories

and tasks. While online tutorials provide abundant knowledge, most existing work focuses on textual tutorials, leaving the rich visual information in videos under-exploited.

**Imitation learning from observations.** Our approach is conceptually related to imitation learning from observations in reinforcement learning, where agents learn from state-only demonstrations without access to expert actions (Liu et al., 2018; Aytar et al., 2018; Yu et al., 2018; Torabi et al., 2019; Guo et al., 2019; Schmeckpeper et al., 2021). In particular, Torabi et al. (2018) has shown that it is possible to train an inverse dynamics model that infers latent actions from state transitions. The trained model is then used to label trajectories in expert demonstrations, which are in turn used for supervised training.

Several recent works have adopted a similar perspective for computer-use agents: tutorial videos provide rich observational demonstrations of how to operate software, but must be converted into actionable steps that agents can execute (Zhang et al., 2025; Jang et al., 2025b; Song et al., 2025; Lu et al., 2025). Specifically, Zhang et al. (2025) crawls online multimodal GUI tutorials (both videos and articles) and converts them into agent trajectories via VLM-based tutorial processing, trajectory generation, and data filtering. The collected trajectories are then used to improve agents by continuous pre-training. Similarly, two concurrent works (Song et al., 2025; Lu et al., 2025) also conduct continuous pre-training from unlabeled videos. To more accurately extract the underlying actions in the video, they further train an inverse dynamics model specialized for the GUI operations.

These works are complementary to ours. They focus on how to accurately label actions and scale up video-derived datasets for offline training, whereas we focus on an inference-time method that does not require additional parameter updates. Given a set of labeled trajectories, our framework studies how to search, filter, and contextualize them so that a computer-use agent can effectively exploit this procedural knowledge on the fly. In addition, our method can potentially be further enhanced if equipped with the specialized inverse dynamics model trained from their works.

## 3 Methodology

### 3.1 Pipeline Overview

In this paper, we aim to develop computer-use agents that automatically complete tasks on the desktop. Given an input task, *e.g.,* "*Fill all the blank cells in B1:E30 with the value in the cell above it*", the initial state, *e.g.,* an Ubuntu desktop with specific files opened, the agent operates on the computer by issuing mouse and keyboard actions. At each step, the agent observes the screenshot of the current desktop and optionally the accessibility tree, and issues the next action by specifying the action type (*e.g.,* click or type) and content (*e.g.,* where to click or what to type). Our goal is to improve the ability of the agent at *inference time* by retrieving relevant online demonstration videos and processing them in a way that allows the agent to learn useful information in context.

To achieve such a goal, we design a pipeline that consists of the following three steps, as illustrated in Figure 1.

1. **Video retrieval:** The agent retrieves and filters relevant online videos that contain useful demonstrations applicable to the testing task.

2. **Video processing:** A VLM extracts underlying actions in the video and summarizes important video segments into demonstration trajectories that contain a description and a sequence of actions and observations.

3. **Video application:** During inference, we design a two-stage selection method to select the most relevant demonstration trajectory at each step. The selected trajectory is provided in context to help the agent better decide the next action.

In the following section, we will describe the details of each step. All prompts used in our pipeline are presented in Appendix A.

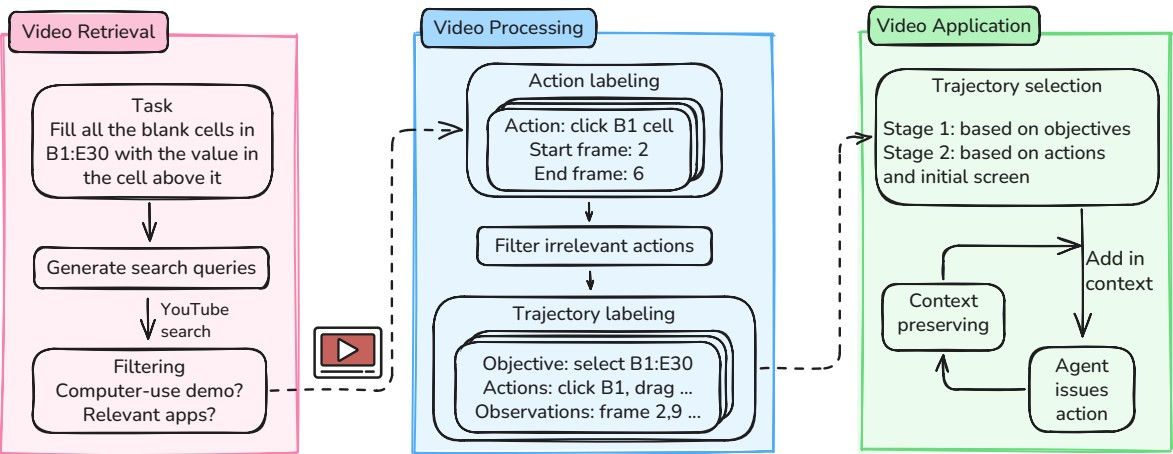

Figure 1: Overview of our inference pipeline. Step 1 outputs a set of videos relevant to the query task. Step 2 yields a set of demonstration trajectories, each with a textual objective and a sequence of actions and observations. Step 3 dynamically selects a single trajectory to add in context before issuing each action, focusing the agent on the most helpful local guidance for its decision.

## 3.2 Video Retrieval

Given a computer-use task described in natural language, our first step is to retrieve online video tutorials that plausibly demonstrate procedures relevant to the task. To achieve this, the agent generates queries for each task, searches for videos using those queries, and filters retrieved videos.

**Retrieving online videos.** We first prompt LLMs to convert the input task into concrete search queries that reflect the application, objects, and intended operation. For the above example, we issue the query: '*LibreOffice Calc, fill blank cells with value from above*'. If the task involves multiple applications, we allow the model to generate one query for each application. We then submit each query to the YouTube API[1] and collect the top-50 results per query. Among the retrieved videos, we only keep those that are shorter than 10 minutes and in English.

**Filtering retrieved videos.** The retrieved videos might be noisy, *e.g.,* a recording of a person speaking without computer operations, or a tutorial on Windows while the task is for Ubuntu, so we filter the retrieved videos to a compact, high-precision set based on LLM assessments.

Specifically, the filtering is done in two steps. ❶ *Coarse selection.* Given the task description, as well as the title and description of all videos, the LLM selects the top-10 videos that are relevant to the task. ❷ *Content verification.* For each kept video, we use a VLM to examine its transcript and a uniform sample of 10 frames spanning the video. The VLM decides whether the video actually shows the application UI and computer operations pertinent to the task. We only keep videos that are deemed helpful.

This stage returns a small set of videos that the VLM has judged as both topically and visually relevant to the task. Multiple videos can be kept for a single task to account for UI and version diversity. A representative result from this stage is a LibreOffice Calc tutorial.[2]

## 3.3 Video Processing

Although the filtered videos contain relevant demonstrations for the task, they are not presented in a way that is ready for use by the agent. Particularly, the video contains continuous frames, but the agent only has

---

[1] https://developers.google.com/youtube/v3.
[2] https://www.youtube.com/watch?v=AeVBH0ClCoA

sparse observations (*i.e.,* screenshots before and after each action). Moreover, the actions performed in the video are not explicitly labeled and must be inferred from screen transitions. To bridge this gap between videos and the agent, we convert each video into structured demonstration trajectories, where each trajectory includes an objective and a sequence of observations and actions.

**Action labeling with a VLM.** The first step of video processing is to label the underlying actions in the video. To make the labeling efficient, we first identify the frames with screen changes and only annotate actions across these frames. Specifically, we downsample each video with 2 frames per second and locate changing frames whose pixel value difference from the previous frame is above a certain threshold. We then prompt a VLM to label the actions in the video. To balance the performance and efficiency, we provide the VLM with a context containing 20 frames with 3 frames of overlap between consecutive context clips. The VLM then extracts ① the UI action type (click, type, right-click, drag, *etc.*), ② the target element (*e.g.,* 'cell B2', 'Find All' button), and ③ start and end frames of the action. Finally, we filter actions based on action types and prompt an LLM to merge the same underlying actions in adjacent clips.

**Filtering important and relevant actions.** The above step is very likely to output actions that are irrelevant to the task, *e.g.,* cursor movement or hovering without any purpose. To improve the quality of labeled actions, we further prompt an LLM to analyze each action and only keep actions that are important and relevant to the main task in the video based on the title, description, and transcript of the video.

**Demonstration trajectory construction.** Even after filtering, a video could contain many actions, which is inefficient for the agent to process. In addition, to determine the next action, the agent usually only needs to focus on a local segment of the video instead of looking at the whole video. To allow the agent to focus on a local segment, we further split the video into subsequences of actions and construct a trajectory for each valid subsequence.

Specifically, we take inspiration from the hindsight relabeling method in RL (Andrychowicz et al., 2017; Su et al., 2025) and re-generate a new objective for a trajectory that does not complete the original task. In particular, given a subsequence of observations and actions, we prompt a VLM to generate a natural language objective that this particular subsequence can accomplish. For example, in our above tutorial on LibreOffice Calc, an objective generated for a segment is '*Search for all occurrences in the spreadsheet using the 'Find All' function in LibreOffice Calc.*' We repeat such a process for all subsequences of actions in the video,[3] and we only keep the results where the VLM deems the subsequence as complete and there exists a reasonable objective. The outcome of this step is a set of *demonstration trajectories*,

---

**Algorithm 1** Trajectory labeling and filtering

**Input:** action-observation sequence $\{(o_t, a_t)\}_{t=1}^N$
**Output:** set of labeled trajectories $\mathcal{T}$
$\mathcal{T} \leftarrow \emptyset$
**for** $i = 1$ **to** $N - 1$ **do**
    **for** $j = i + 1$ **to** $\min(N, i + 15)$ **do**
        Use VLM to generate an objective $g_{i:j}$ for the action sequence $a_{i:j}$
        **if** the sequence is complete and coherent **then**
            $\mathcal{T} \leftarrow \mathcal{T} \cup \left\{ \left( g_{i:j}, \{(o_t, a_t)\}_{t=i}^j \right) \right\}$
        **end if**
    **end for**
**end for**
**return** $\mathcal{T}$

---

represented as $\left\{ \left( \text{objective}, \{(o_t, a_t)\}_{t=1}^{l_i} \right) \right\}_{i=1}^K$, where $o_t$ is the screenshot at step $t$, $a_t$ is the corresponding action, $l_i$ is the length of $i$-th trajectory, and $K$ is the total number of trajectories in the video.

The VLM might generate sub-optimal objectives for the trajectory, where the action sequence does not fully complete the objective. To ensure high quality of constructed trajectories, we further prompt a VLM to filter out the invalid trajectories. Particularly, the VLM examines the coherence and correctness of each constructed trajectory, and trajectories failing the check are discarded. Algorithm 1 details the labeling and filtering process for trajectory construction.

---

[3]For efficiency, we only consider subsequences with length less than or equal to 15.

Appendix C includes an example walking through the video processing pipeline. It illustrates how our pipeline identifies changing frames, labels actions, generates an objective that matches the task requirement, and filters invalid trajectories.

### 3.4 Video Application

With the processed videos, at inference time, the agent dynamically selects zero or one trajectory at each step to better decide the next action. We develop the following two-stage selection framework, which allows the agent to more accurately select a trajectory to focus on.

❶ *Stage 1: Coarse selection.* Given the current observation and task, the agent selects a candidate pool of trajectories from all videos of the task. Specifically, the VLM reads each trajectory's objective and selects the top-3 relevant trajectories for each video. Since the input only contains the textual objectives without the observation and action sequence, the selection is efficient. ❷ *Stage 2: Detailed selection.* The agent then inspects the candidate pool more closely by additionally checking the initial observation and the action sequence of each trajectory. It selects a single trajectory that is most helpful for determining the next action, or no trajectory if all candidates are irrelevant. Finally, the selected trajectory, including the objective, observation, and action sequence, is provided in context when the agent generates the next action.

**Maintaining coherent context across steps.** Selecting trajectories at each step could lead to frequent context changes for the agent across consecutive steps, which might affect performance (*e.g.,* when the agent selects a trajectory and starts to follow, it could select a different trajectory at the next step, which deviates from the previous context and plan). To minimize unnecessary reselection, at each step, the agent first decides if the previously selected trajectory is still relevant and applicable for the current observation. If yes, we simply continue with the same trajectory. Otherwise, we repeat the two-stage selection to obtain a new trajectory. This keeps the in-context guidance stable when progress is steady, while allowing quick adaptation when the UI diverges.

## 4 Experiments

In this section, we conduct experiments to verify the effectiveness of our proposed method. Particularly, we aim to answer two questions: ① Does providing video tutorials improve agents' performance on computer-use tasks? ② What factors affect the agents' ability to learn from videos?

### 4.1 Experimental Setting

**Evaluation benchmarks.** We evaluate on two widely used benchmarks for computer-use tasks. ❶ OSWORLD-VERIFIED (Xie et al., 2024; 2025c) (hereafter as OSWORLD) contains 369 Ubuntu desktop tasks spanning real applications such as GIMP and VSCode, as well as tasks that involve multiple applications. Each task is associated with a reproducible initial state and manually designed evaluators. Following official guidance, we exclude 8 Google Drive tasks for ease of deployment. ❷ WEBARENA (Zhou et al., 2024) contains 812 tasks focusing on the web environment across different domains, such as GitLab and shopping. The agent needs to navigate the browser to complete a given task. As the official server for the map domain is not accessible, we evaluate on the remaining domains. For both benchmarks, we use the official evaluation implementation and report the final success rate (SR) over all tasks.

**Baselines.** We compare with three strong baselines. ❶ **Base agent**: We adopt the state-of-the-art agent frameworks on both benchmarks. These frameworks do not have access to external tutorials during inference. Specifically, on OSWORLD, we use `Jedi` (Xie et al., 2025b) as the base agent; on WEBARENA, we use `AgentOccam` (Yang et al., 2025b). ❷ **Text-only tutorials:** We retrieve online textual tutorials relevant to the task and prepend the retrieved tutorials to the context at each step. For retrieval, we follow Agent S (Agashe et al., 2025a) to first generate queries for each task and then use Perplexica Search Engine[4] to retrieve and summarize online tutorials into a comprehensive document. ❸ **Video transcript only:** We

---

[4] https://github.com/ItzCrazyKns/Perplexica.

Table 2: Success rate on WEBARENA.

|  | Shopping | Shopping Admin | Reddit | GitLab | All |
|---|---|---|---|---|---|
| AgentOccam (Yang et al., 2025b) | 44.8 | 56.1 | 60.2 | 33.9 | 47.3 |
| + Transcript | 44.6 | 54.2 | 53.6 | 42.0 | 47.9 |
| Ours | **47.1** (+2.3) | **58.5** (+2.4) | **60.6** (+0.4) | **44.1** (+10.2) | **51.5** (+4.2) |

retrieve and filter online videos as described in Section 3.2, but we only use their transcripts without the visual information. To obtain the transcript, we use the `turbo` model from Whisper (Radford et al., 2023). Note that baselines ❷ and ❸ are built upon the base agent frameworks in ❶ by providing external knowledge.

**Agent settings.**  For all baselines and our method, we use `gpt-5-mini-2025-08-07` as the underlying VLM backbone at inference. On OSWORLD, all methods (ours and baselines) receive *only* the current screenshot as the observation, following prior work Xie et al. (2025b). We also set the maximum number of steps to 50. On WEBARENA, we follow Yang et al. (2025b) to use the accessibility tree of the page, as well as the additional current screenshot as the observation. The maximum steps is set to 20. Additional parameters of the agents are detailed in Appendix A.

**Video sources.**  On OSWORLD, we directly retrieve videos from YouTube, as described in Section 3.2. After filtering, we collect videos for 211 out of 361 tasks,[5] and on average, each task contains 3.6 videos.[6] On WEBARENA, we use the videos from VideoWebArena (Jang et al., 2025a), which contains pre-recorded videos for 672 tasks in WEBARENA, excluding the tasks that require the map server. Note that although these videos are specifically designed to cover the skills required in WEBARENA, they do not align with the evaluation tasks exactly, and one video could contain multiple skills corresponding to many tasks. Thus, the agent still needs to select relevant trajectories from each video and adapt its content to tasks at hand, instead of directly copying the video.

**Implementation details.** We build our method based on `Jedi` and `AgentOccam` on the two benchmarks, respectively. For video retrieval and processing, we use `Qwen2.5-VL-32B-Instruct` (Bai et al., 2025) and `Qwen3-30B-A3B-Instruct-2507` (Yang et al., 2025a) as the VLM and LLM, respectively. Additional implementation details are in Appendix A.

## 4.2  Main Results

**Overall success rate.**  Table 1 shows the results on OSWORLD. We separately report the performance on the subset of 211 tasks for which we retrieve at least one relevant video. On the remaining 150 tasks, our method and the transcript baseline behave exactly the same as the `Jedi` framework. As the table indicates, our method outperforms all three baselines, especially on the subset that has video tutorials. This result confirms that our

Table 1: Success rate on OSWORLD. "Has Videos" denotes the subset of 211 tasks for which we find at least one relevant video.

|  | Has Videos (211) | All (361) |
|---|---|---|
| Jedi (Xie et al., 2025b) | 46.8 | 41.0 |
| + Text tutorial | 44.0 | 39.0 |
| + Transcript | 46.7 | 41.0 |
| + Video demo | **50.3** (+3.5) | **43.1** (+2.1) |

proposed pipeline improves the agent's ability by learning from videos in context. Particularly, providing transcripts only does not improve the performance, which suggests that our structured, visually grounded demonstration trajectories are more useful than unstructured textual guidance during inference. Additionally, on average, our method takes 28.8 steps to complete a task, which is fewer than that of the `Jedi` baseline (30.3). This suggests that having access to external knowledge allows the agent to finish the tasks more directly without unnecessary exploration.

---

[5]The remaining 150 tasks do not have a relevant video. On those tasks, our method behaves the same as the baseline agent.

[6]The number is calculated on the subset of 211 tasks, and these are the videos after our filtering process.

Table 2 shows the results on WEBARENA across different domains. Since the websites are synthetically built and do not exist in the real world, we do not include the text tutorial baseline. As can be observed, the results show similar trends as in Table 1, where our method consistently achieves better performance than the baseline framework and the variant that adds transcript information. Particularly, in more challenging domains like GitLab, our method achieves a more significant improvement (+10.2), indicating a good utilization of the videos. By contrast, in easier domains like Reddit, the agent already has enough knowledge of how to complete the task (*e.g.,* how to create a post or make a comment), therefore adding additional information in the video does not make a big impact.

### 4.3 Additional Analyses

**Computational costs.** Section 4.2 shows that incorporating video demonstrations improves task success, but it also introduces extra computation from processing videos and selecting relevant trajectories at each step. We now quantify this overhead and discuss potential optimization. Specifically, we measure the additional cost for the three stages of our method on a random subset of 100 tasks on OSWORLD. Table 8 summarizes the latency for each step. Overall, video retrieval contributes minimal overhead, as it involves relatively lightweight operations. The dominant cost in video processing comes from trajectory construction, since it needs to iterate over subsequences of actions to generate objectives. In the final video application stage, trajectory selection increases per-task runtime by 48.9% compared to the baseline that uses no video information. We view this overhead as acceptable given the corresponding performance gains and the fact that our approach is inference-time (*i.e.,* it does not require parameter updates or additional training data). In practice, our method is most attractive when users can afford roughly a $\sim 1.5\times$ inference-time cost, or when training is infeasible due to data, compute, or access constraints.

Additionally, several implementation choices can be introduced to further reduce the overhead. First, the first two stages can be done offline, so that the actual user inference only pays the trajectory selection cost.[7] Second, the videos can be processed at a lower frame rate (currently 2 frames per second) when identifying changing frames and labeling actions, so that both the number of actions and constructed trajectories will be reduced. Third, during execution, the agent can self-assess whether it already has sufficient knowledge to decide the next action; if so, it can skip trajectory selection and proceed using its internal knowledge.

**Failure cases.** To inspect the failure mode of our method, we manually examine 20 tasks in OSWORLD where the baseline agent succeeds, but our method fails. Appendix B lists the categories of failures and shows concrete examples. Overall, we observe that a majority of the failure cases are not caused by adding video information, and among video-related failures, the main causes are: ① video retrieval, where the system cannot find useful video tutorials; and ② video application, where the agent cannot adapt when the UI version in the video tutorial is different from its environment. This suggests that future improvements can be made by improving the agent's ability to adapt the selected trajectory to the testing environment.

**Robustness to noisy videos.** We next study the robustness of our method to noisy videos. Specifically, instead of using videos after careful filtering in Section 3.2, we take the videos that do not pass the first-step coarse selection of our filtering process. These videos are usually irrelevant to the task (*e.g.,* for the task *"Could you help me stretch this image to fill the entire page, keeping its proportion and centering the image?"* on LibreOffice Impress, one video title is *"How to insert picture into table in word"*). We test our method's performance when using these noisy videos while keeping everything else the same. On a subset of 50 tasks in OSWORLD, we observe that using noisy videos actually solves one more task than using filtered videos. Together with the error analysis in Figure 2, where the agent is misled by a video showing a similar task but in a mismatching UI version, the results show that our system is more robust to clearly irrelevant videos, but it is more likely to be affected by the adversarial videos that look similar but contain misleading guidance.

---

[7]Following Zhang et al. (2025), one can synthesize a set of seed task queries and use them to retrieve, filter, and process videos before receiving user queries. When a user query arrives, we can retrieve similar synthetic queries and use the corresponding already-processed videos for inference.

### 4.4 Ablation Study

We now investigate how different factors in our pipeline affect the final performance. Specifically, we analyze the impacts from four aspects, including the number of videos per task, the proposed two-stage trajectory selection method, the filtering step in video processing, and the importance of visual information in the video. We conduct all the following analyses on the subset of 211 tasks in OSWORLD that have at least one retrieved video.

**Number of videos per task.** We first study how the number of videos per task affects the performance. Specifically, during inference, we restrict the agent's access to only a subset of retrieved and filtered videos. Table 3 shows the performance of our method as we vary the amount of available videos. As can be seen, having access to more videos improves the overall performance, since the agent can select the demonstration trajectory from a larger pool. This suggests that **our method has the potential to further improve performance as more relevant videos become available, assuming retrieval quality and selection remain effective**.

Table 3: Performance on OSWORLD with different numbers of videos per task.

|  | Success Rate |
|---|---|
| 1 video per task | 46.9 |
| 3.6 videos per task | 50.3 |

**The two-stage trajectory selection.** We then explore the effect of our dynamic two-stage trajectory selection method, which allows the agent to focus on a local segment of the video at each step. Specifically, we compare with a variant of our method that disables this mechanism: instead of constructing trajectories for all subsequences of actions and dynamically selecting a trajectory at each step during inference, we always prepend the longest trajectory that corresponds to the entire video. If a task has multiple videos, we first prompt the agent to select the most relevant one and always provide that video during inference. The second row of Table 4 (no trajectory selection) shows that the performance of this variant drops significantly compared to our original method. This indicates the **importance of splitting the video into short trajectories and dynamically selecting suitable trajectories at each step**.

Table 4: Ablation study on OSWORLD.

|  | Success Rate |
|---|---|
| Ours | 50.3 |
| No trajectory selection | 45.8 |
| No action filtering | 46.6 |
| No visual information | 45.6 |

**Action filtering in video processing.** Next, we investigate the impact of filtering actions during video processing in Section 3.3. Recall that after labeling actions in the video, we filter actions and only keep those that are relevant to the main task of the video. To validate the impact of this filtering step, we compare with a variant of our method without this step, while keeping everything else the same, including the trajectory selection method. Results in Table 4 show that this variant (no action filtering) is worse than our original method, which illustrates the **benefits of filtering labeled actions**.

**Visual information in trajectories.** Finally, we study whether the visual information in trajectories contributes to the final performance. Specifically, we modify our method so that when providing the selected trajectory in context, we only provide the textual information (objective and action sequence), without the screenshot at each step. Table 4 shows that this variant (no visual information) is worse than our original method, which provides the sequence of screenshots in context. This indicates that **text-only summaries cannot cover all useful information in the video**, and adding visual information can further boost the performance of the agent.

## 5 Conclusion

We studied how computer-use agents can learn from online videos at inference time, inspired by how humans watch and imitate short segments of tutorials to acquire domain-specific procedural knowledge. We proposed

a framework that retrieves and filters tutorial videos, converts them into structured, visually grounded demonstration trajectories using a VLM, and dynamically selects a single trajectory as in-context guidance at each step. Experiments show consistent gains over strong base agents and variants using only textual tutorials or transcripts, and analyses highlight the importance of trajectory segmentation and selection, action filtering, and visual information.

## 6  Limitations

Despite the performance improvements, our method has two limitations that may affect its practicality.

**Inference-time overhead.** Our approach introduces additional inference-time cost from video processing and per-step trajectory selection. As shown in Section 4.3, the final application stage increases per-task runtime by 48.9% compared to a baseline agent without video guidance. While this overhead is moderate relative to the observed gains, it may be undesirable in latency-sensitive deployment settings. As a result, our method is best suited for scenarios where higher task reliability is prioritized over minimal response time, or where training is impractical due to limited data or compute access. More generally, this overhead reflects a trade-off between inference efficiency and leveraging external procedural knowledge at test time.

**Sensitivity to UI mismatch and near-miss demonstrations.** Although our system is generally robust to irrelevant videos, it remains sensitive to videos that appear highly relevant but differ subtly from the execution environment due to software version changes or alternative UI layouts. In these cases, the agent may over-rely on the selected trajectory and fail to adapt it to the current observation, leading to execution errors. This limitation suggests that stronger mechanisms for grounding demonstrated actions in the current environment are important directions for future work.

## 7  Broader Impact

This work studies how computer agents can acquire procedural knowledge at inference time by learning from publicly available tutorial videos. While this direction has the potential to substantially improve agent adaptability and usability, it also raises important considerations, which we discuss below.

**Copyright and Terms of Service.** Our system retrieves and processes publicly available tutorial videos from YouTube. We do not redistribute raw videos. Instead, the system operates on limited derived information, including transcripts and sparsely sampled frames. All processed artifacts are used for research purposes only and are not intended to replicate the original content. We design the pipeline to comply with the platform's terms of service and do not enable downloading or redistribution of copyrighted videos.

**Security and Safety Risks.** Tutorial videos may demonstrate operations that are unsafe, irreversible, or undesirable if blindly executed by the agent. In our experiments, agents operate in sandboxed virtual environments, which limits real-world risk. Nevertheless, this risk highlights the importance of incorporating additional safety mechanisms, such as restricting action spaces, filtering or flagging potentially dangerous operations, and requiring agent self-verification before executing irreversible actions. We therefore recommend that users adopt this system with appropriate caution and safety safeguards.

**Bias and Representational Concerns.** In this work, we restrict our experiments to Ubuntu-based environments on OSWORLD and therefore retrieve and process only Ubuntu-related tutorials. On WEBARENA, we use the officially built browser environment. Generalization to other operating systems or substantially different interfaces is not evaluated here and remains an important direction for future work.

**Privacy Considerations.** Our approach relies exclusively on publicly accessible content and does not access private user data. The agent does not store or infer personal information beyond what is explicitly contained in public tutorials.

## 8 Acknowledgements

The work of Yujian Liu and Shiyu Chang was partially supported by National Science Foundation(NSF) Grant IIS-2338252, NSF Grant IIS-2207052, NSF Grant IIS-2302730, and the Open Philanthropy Research Award.

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

Table 5: List of prompts in our pipeline.

|  |  | **Prompt** |
|---|---|---|
| Video Retrieval | Query generation | Figure 11 |
|  | Video filtering step 1 | Figure 12 |
|  | Video filtering step 2 | Figure 13 |
| Video Processing | Action labeling | Figure 14 |
|  | Action merging | Figure 15 |
|  | Action filtering | Figure 16 |
|  | Trajectory objective generation | Figure 17 |
|  | Trajectory filtering | Figure 18 |
| Video Application | OSWORLD Trajectory selection stage 1 | Figure 19 |
|  | OSWORLD Trajectory selection stage 2 | Figure 20 |
|  | WEBARENA Trajectory selection stage 1 | Figure 21 |
|  | WEBARENA Trajectory selection stage 2 | Figure 22 |
|  | Trajectory continuation | Figure 23 |

Table 6: Agent setting on OSWORLD.

|  | Value |
|---|---|
| Observation space | Screenshot |
| Maximum steps | 50 |
| Reasoning effort for VLM | medium |
| Temperature | Default |

Table 7: Agent setting on WEBARENA.

|  | Value |
|---|---|
| Observation space | Screenshot & accessibility tree |
| Maximum steps | 20 |
| Reasoning effort for VLM | low |
| Temperature | Default |

# A  Implementation Details

## A.1  Video Retrieval and Processing

We follow the procedure described in Sections 3.2 and 3.3 to retrieve and process online videos. We use `Qwen2.5-VL-32B-Instruct` (Bai et al., 2025) and `Qwen3-30B-A3B-Instruct-2507` (Yang et al., 2025a) as the VLM and LLM, respectively. Table 5 summarizes the detailed prompts we use in our pipeline. Note that the video retrieval step is only done for OSWORLD. On WEBARENA, we use the pre-recorded videos from Jang et al. (2025a). Additionally, since each video in Jang et al. (2025a) covers knowledge for multiple tasks, we do not filter labeled actions on WEBARENA. Instead, we directly construct demonstration trajectories for all subsequences of actions.

In particular, to construct demonstration trajectories in the video processing stage, we iterate over all subsequences of actions that have a length shorter than or equal to 15. For each subsequence of actions, we provide the VLM with all actions (in text), as well as the beginning and end frames of the subsequence (we omit the intermediate frames for efficiency). The VLM is instructed with the prompt in Figure 17 to generate an objective for this subsequence, or None if the actions do not achieve any reasonable task. We then filter the labeled trajectories using the prompt in Figure 18 to make sure the final trajectories are coherent and complete. Because we limit each trajectory to at most 15 actions, the complexity of this step reduces from $O(N^2)$ to $O(N)$. Additionally, since we only provide the VLM with start and end frames without intermediate frames, the latency for each VLM query is relatively small, making it scalable to videos with hundreds of actions.

### A.2 Agent Framework

Our method is built on the `Jedi` (Xie et al., 2025b) and `AgentOccam` (Yang et al., 2025b) frameworks, respectively. We use `gpt-5-mini-2025-08-07` as the underlying VLM backbone for all methods. Table 5 shows the prompts we use for the trajectory selection at each step. Tables 6 and 7 summarize the agent parameters for OSWORLD and WEBARENA, respectively.

## B Analysis of Failure Cases

To analyze the behavior of the agent, we conduct an error analysis for our method. Particularly, we manually examine 20 tasks in OSWORLD where the baseline agent (`Jedi`) is correct but our method is wrong. We categorize the error causes into two groups: one group that is related to our video pipeline and the other that is not.

For the errors that can be attributed to our video pipeline (6/20), we identify the following reasons:

- Mismatch between the video's UI version and the agent's UI version (1/20). We identify a case where the video shows how to click "Display Google Chrome in this language" in order to change the default language, but the agent is operating on a Chrome in a different version that does not have the same button. This discrepancy led to confusion and repetitive clicking of the same button (Figure 2).

- The agent already completed the task but followed the video to do an extra step (2/20). For example, for a task that requires searching for specific flights. The agent already applied the required constraints and obtained the search results, but it clicked on one of the flights for subsequent purchase (Figure 3).

- The agent did not select the useful trajectory or did not find useful videos (2/20). For a task that requires showing the power percentage, although the video processing pipeline has constructed a trajectory demonstrating how to achieve that in settings, the trajectory is not selected during inference. In another task that requires exporting the address book from Thunderbird and converting it to a xlsx file, our framework did not find any video tutorial on exporting in Thunderbird and only retrieved a video about using LibreOffice to convert a file to xlsx. In both cases, the agent lacks useful guidance and fails the task.

- The agent did not follow the selected trajectory for the next action (1/20). In a task that requires copying a table and pasting the transposed table, the agent already selected a trajectory demonstrating the procedure to complete the task, but it did not follow the trajectory to click the buttons and instead used a hotkey, which led to a wrong outcome (Figure 4).

For the errors that are not related to our video pipeline (14/20), we identify three main groups:

- Grounding issues (6/20). In these cases, the agent planned the correct next action (*e.g.,* scroll down the list to find a specific item or drag an element to a specific position). However, to actually execute these actions, it needs to generate code that specifies the precise amount for scrolling and the precise (x, y) coordinates for dragging and clicking. When the coordinates are wrong, the actions will not be successfully executed, and the agent keeps repeating the same actions (Figure 5). This is also noted as a major failure case in the original OSWORLD benchmark Xie et al. (2024).

- Other issues (6/20). These failures are caused by unrelated system errors, such as network connection errors, or the agent is distracted by pop-up windows while browsing the web.

- Evaluation errors (2/20). In two tasks, the agent correctly completed the task (*e.g.,* flip an image horizontally), but the evaluation was judged as incorrect.

In summary, we observe that a majority of the failure cases are not caused by our method that adds the video information. Among the failures that are related to our method, most errors are from ① video retrieval,

where the system could not find useful videos; and ② video application, where the agent could not adapt when the UI version in the video is different from what it sees, or it follows the videos to do extra actions that are not required in the tasks. This suggests that future improvements can be made by improving the agent's ability to adapt the selected trajectory to the testing task.

## C    Example of Video Processing

In this section, we provide a concrete example showing our video processing pipeline. The task is "*Please help me set Chrome to delete my browsing data automatically every time I close the browser*", and a video our system retrieved is `https://www.youtube.com/watch?v=v0kxqB7Xa6I`. Our video processing consists of the following five steps:

1. **Changing frames detection.** Given a video, we sample frames at 2 frames per second and identify frames that are significantly different from the previous frames, based on the absolute pixel difference. These identified frames represent candidate timestamps where an action may happen. Figure 6 shows two examples.

2. **Action labeling.** We provide a sequence of 20 frames from the last step to the VLM. The VLM is instructed to label actions in the sequence. For each labeled action, it also needs to specify the start and end frames of the action. For example, if a click action involves moving the mouse, clicking the button, and waiting for the page to load, then the start frame is before the mouse movement, and the end frame is after the page loading. The intermediate frames of an action will be ignored in later steps. Figure 7 shows two examples. Note that to label the entire video, we split the video into chunks of 20 frames, and keep 3 overlapping frames between consecutive chunks to maintain the context.

3. **Acton filtering.** Given the labeled actions, we merge the duplicate actions at the boundary of two chunks and remove irrelevant actions.

4. **Trajectory objective generation.** We iterate over all subsequences of actions with length shorter than or equal to 15. For each subsequence, we provide the VLM with the action list, as well as the start and end frames of the subsequence. The VLM is instructed to generate an objective for the subsequence, or None if the actions do not complete a reasonable task. Figure 8 shows an example.

5. **Trajectory filtering.** Given labeled trajectories, the system filters and removes trajectories that are incomplete, incoherent, or contain unnecessary steps. The resulting trajectories are used during task inference, so that the agent can select the most suitable trajectory at each step. Figure 9 shows an example of the final structured trajectory that is provided to our agent. Figure 10 shows an example of a filtered trajectory.

## D    Detailed Computational Cost

We measure the latency on a random subset of 100 tasks on OSWORLD. For a fair comparison, we run our method and the baseline (`Jedi`) with the same configuration during the same time period (to control for OpenAI API latency variations). For video retrieval and processing, we serve the LLM/VLM with vLLM Kwon et al. (2023) on 8 AMD MI250 GPUs. Table 8 presents the results.

**Task**: I am more familiar with Korean as I am from Korea. I want to use chrome with my mother tongue. Could you help me change the Chrome interface language to Korean?

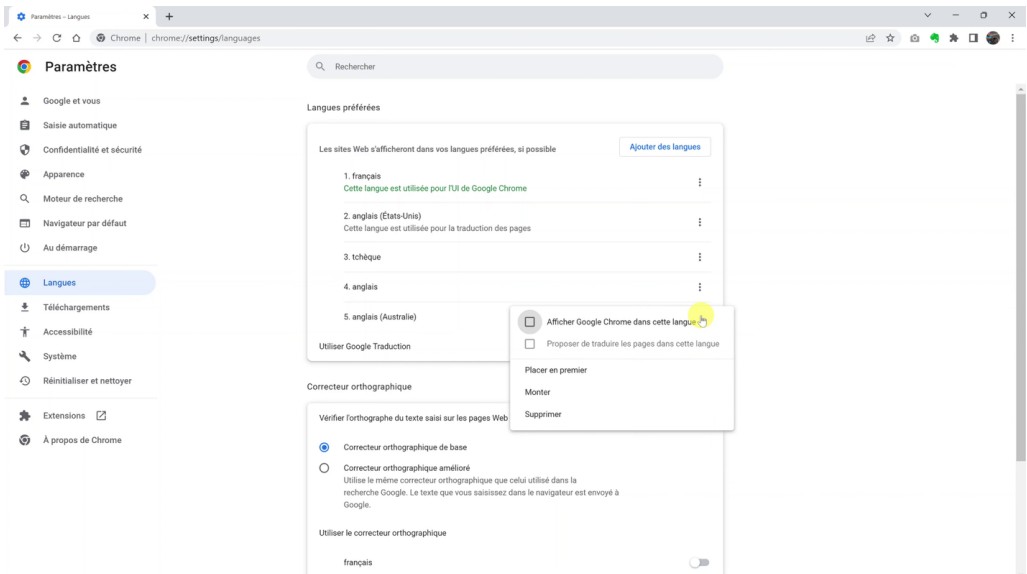

A frame in demonstration

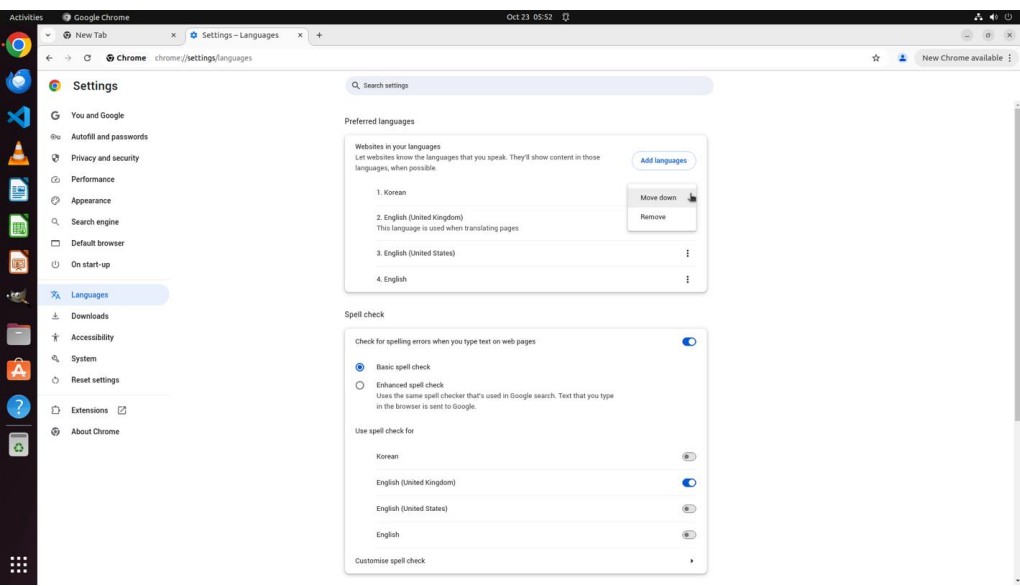

Agent's environment

Figure 2: An example that shows the UI mismatch between the video tutorial and the agent's environment. The tutorial demonstrates how to set English as the default language in Chrome by clicking the "Display Google Chrome in this language" button (translated). However, in the agent's environment, there is no such button, which leads to the task failure.

**Task**: Find flights from Seattle to New York on 5th next month and only show those that can be purchased with miles.

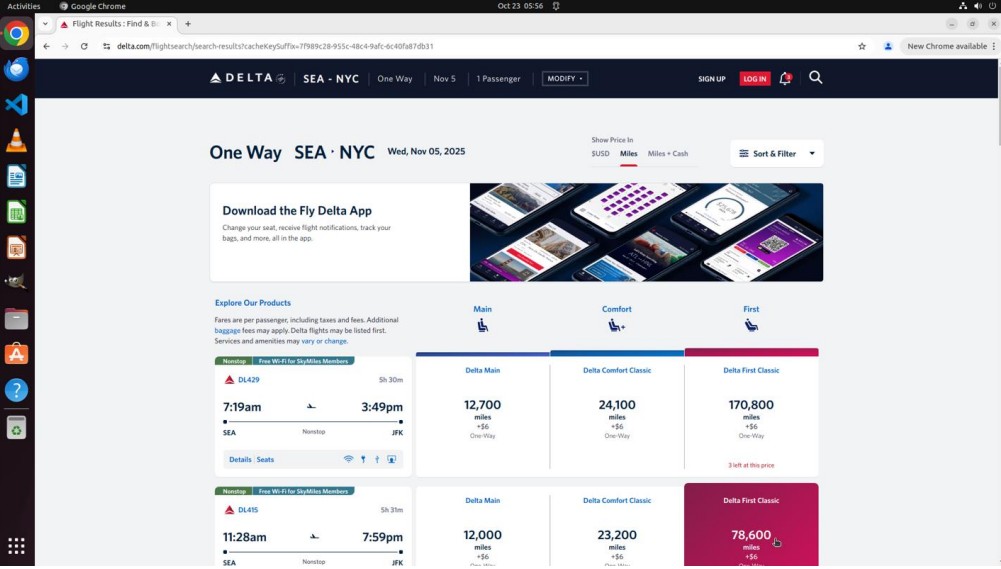

At this step, the agent has correctly searched for flights satisfying the task, but it proceeded by clicking and purchasing the first flight.

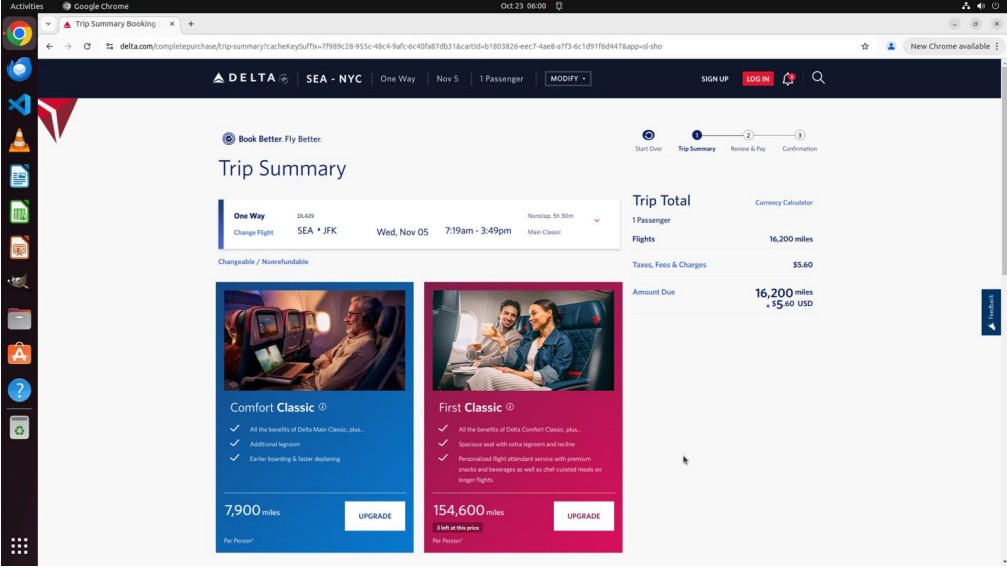

The agent finally ended at the purchase page of a specific flight, which failed the evaluation.

Figure 3: An example that shows the agent has correctly completed the task by searching the flights, but it followed the demonstration to click on a specific flight.

**Task**: Apply matrix transposition to the table in B2:F5 and paste the transposed table at B8 (i.e., the top-left cell of the transposed table should be at B8)

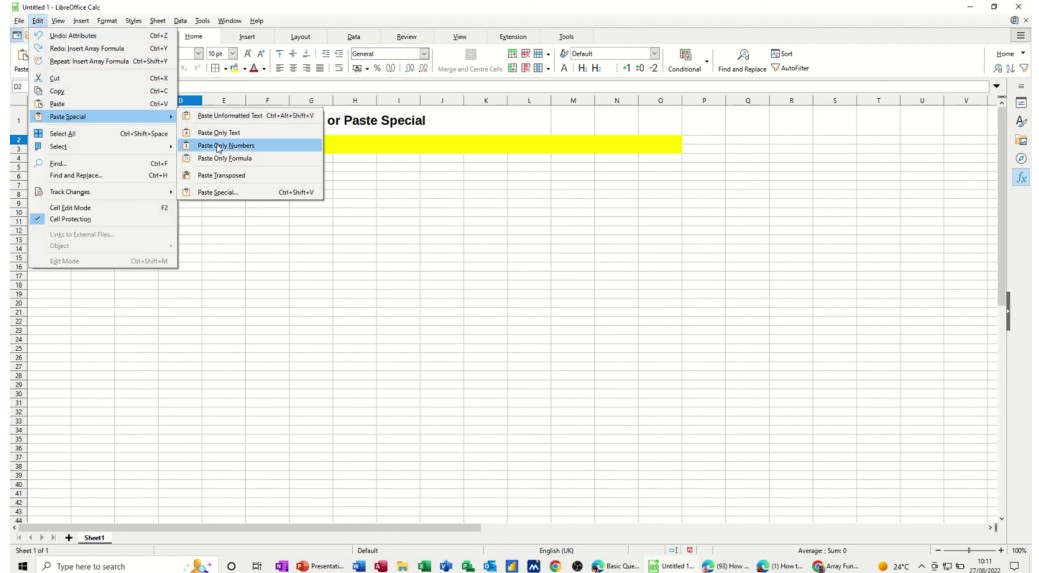

A frame in demonstration. In this demonstration, the labeled actions are (1) click the [Paste Special] option, and (2) click the [Paste Transposed] option.

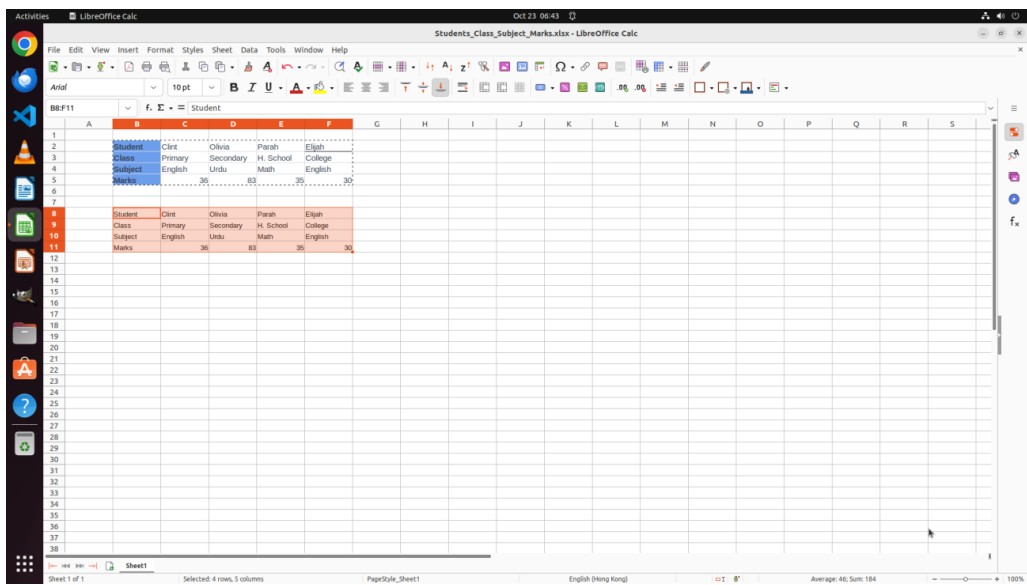

Agent's environment. Although the demonstration clearly shows how to paste the transposed table, the agent did not follow and instead used hotkey for pasting, which led to a wrong outcome.

Figure 4: An example that shows the agent has selected a useful demonstration trajectory, but it did not follow the demonstration and ended in failure.

**Task**: I need to set the decimal separator as a comma (,) for localized data representation and clarity in visualization. Can you help me to update all the numbers in the sheet? Also please keep the decimal numbers as-is.

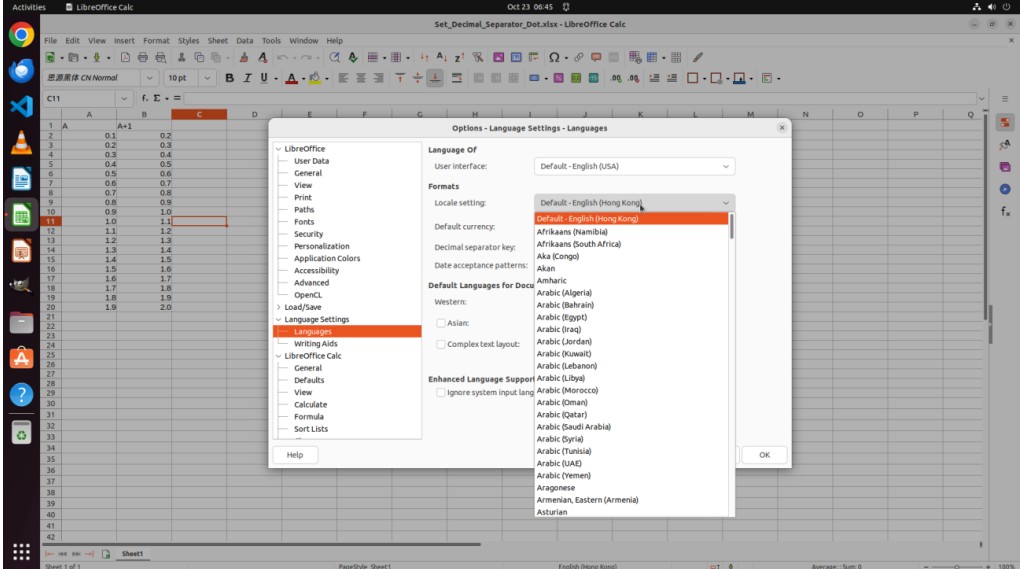

At this step, the agent correctly planned to select German, which uses comma as separator, but it failed to scroll the list to find German.

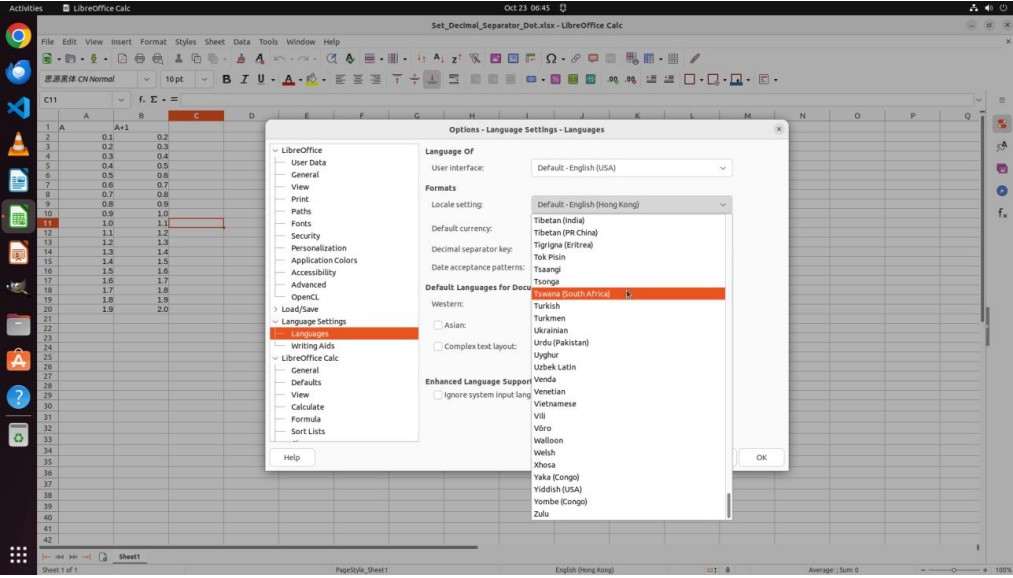

After scrolling, it missed German, and the agent repeated the scrolling back and forth.

Figure 5: An example that shows the agent failed at grounding the correctly planned action to the precise operation command.

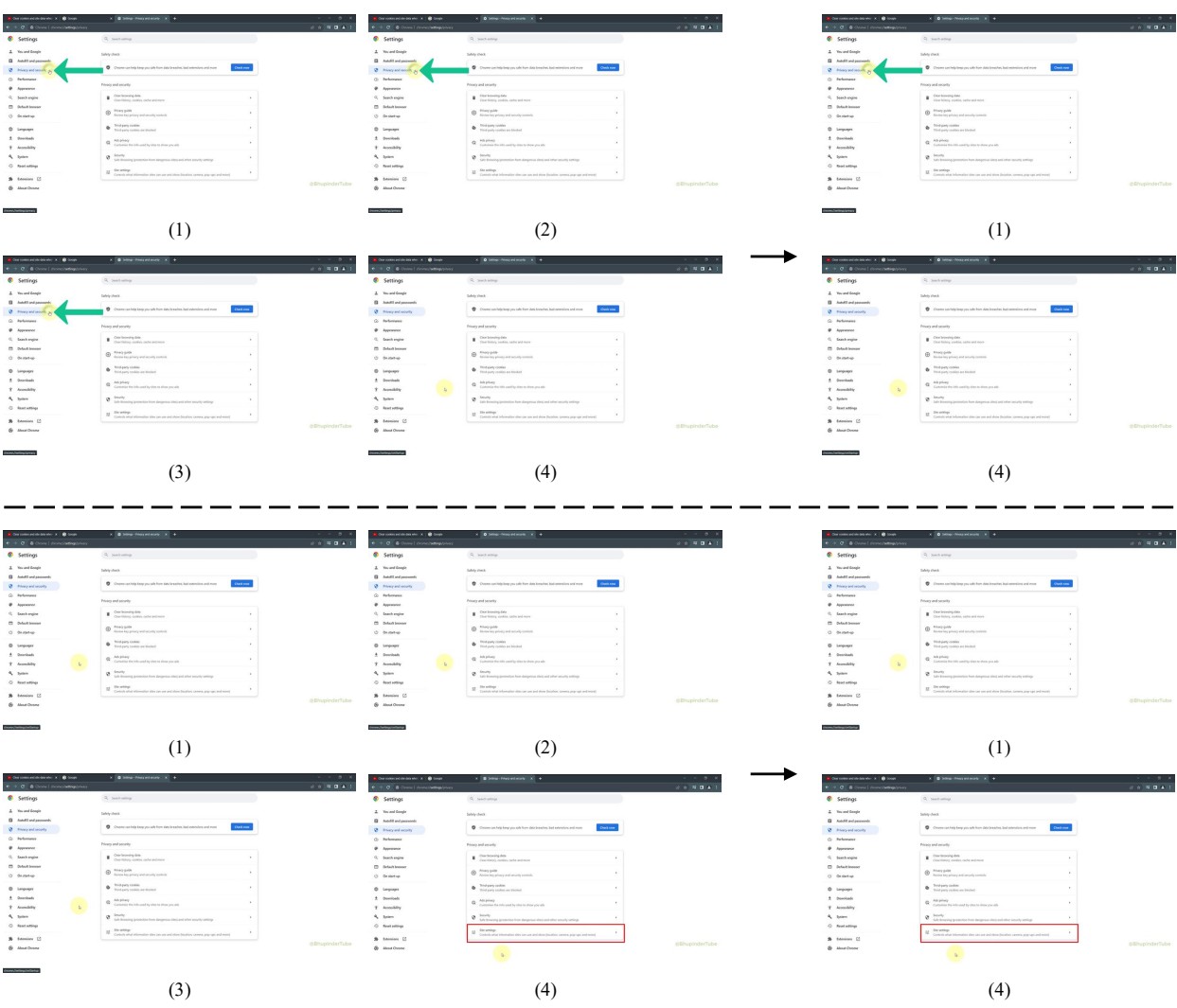

Figure 6: Two examples that show how our system identifies frames that have significant changes from the previous frames. The resulting frames are candidate timestamps where an action may happen, and they are provided as input to the next step of action labeling.

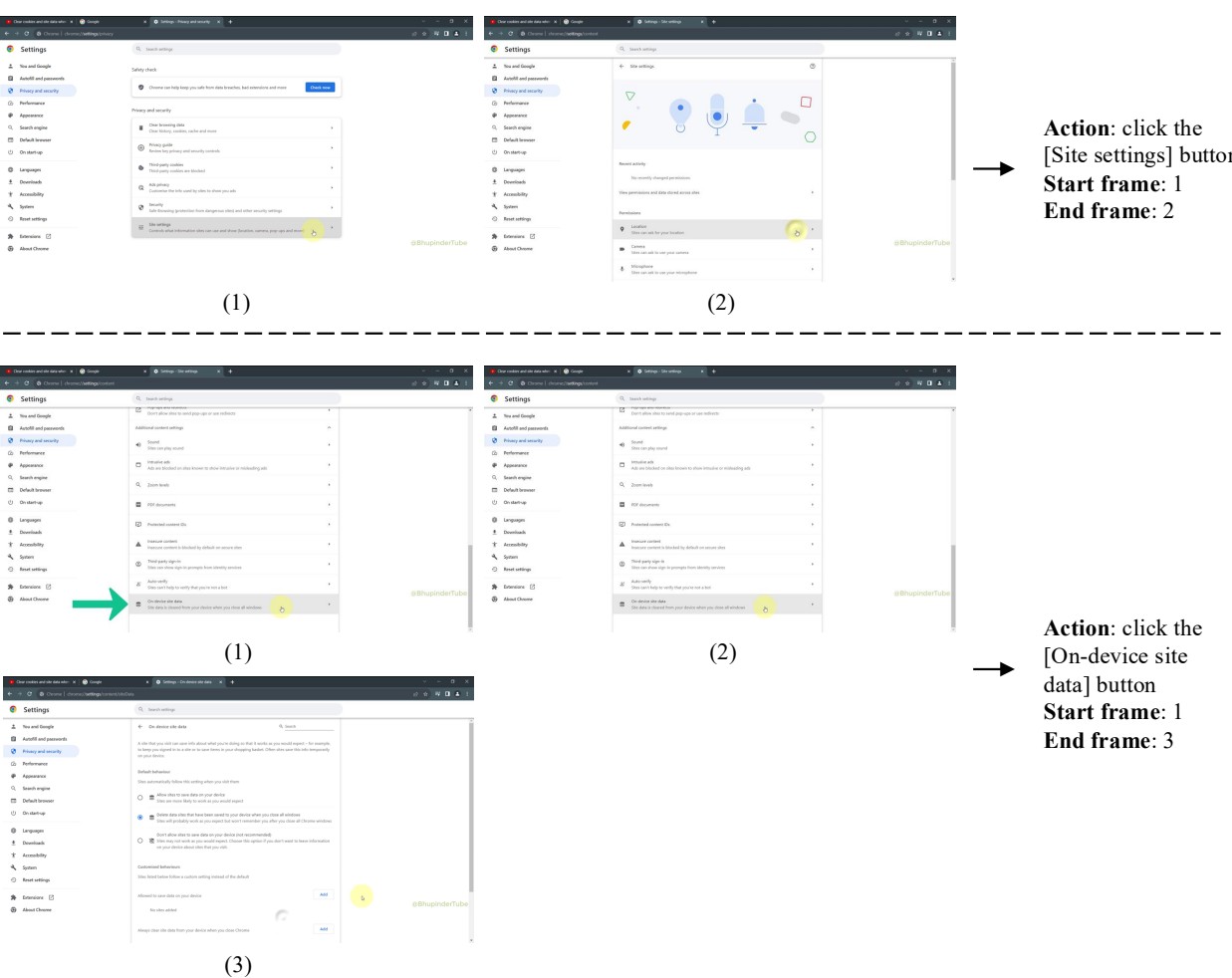

Figure 7: Two examples that show how our system labels actions from a sequence of frames. Note that in the actual implementation, we pass a sequence of 20 frames to the VLM, but we only show the relevant frames here for ease of visualization. Each action is labeled with associated start and end frames, which indicate the start and completion point of the action. The intermediate frames will be ignored for later processing.

**Action sequence:**
1. click the [Privacy and security] button
2. click the [Site settings] button
3. click the [Additional content settings] button
4. scroll down the settings menu
5. click the [On-device site data] button
6. highlight the [Delete data sites that have been saved to your device when you close all windows] option

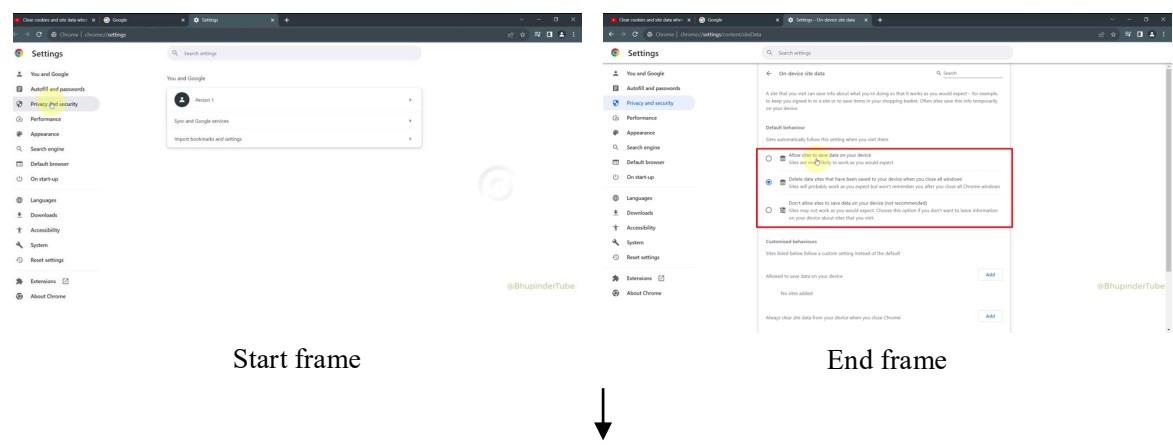

Start frame            End frame

**Objective**: Configure Chrome to delete site data when all windows are closed.

Figure 8: An example that shows how our system generates an objective for a sequence of actions. For efficiency, we limit the action sequence to at most 15 actions, and we only provide the start and end frames of the sequence to the VLM.

**Objective**: Configure Chrome to delete site data when all windows are closed.

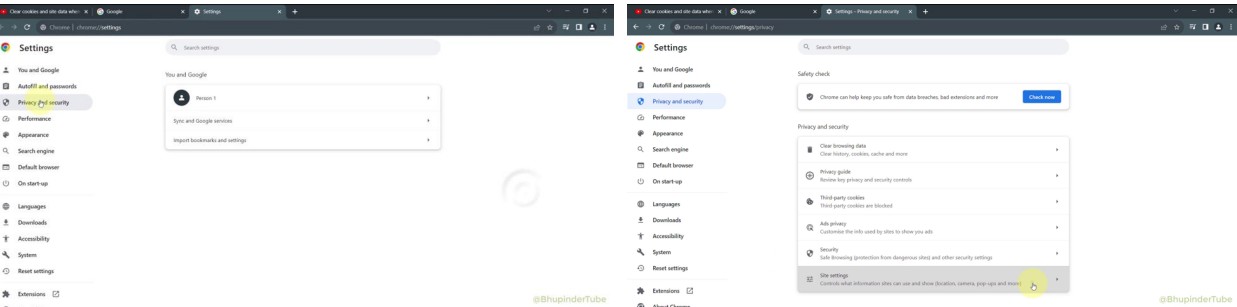

Action 1: click the [Privacy and security] button          Action 2: click the [Site settings] button

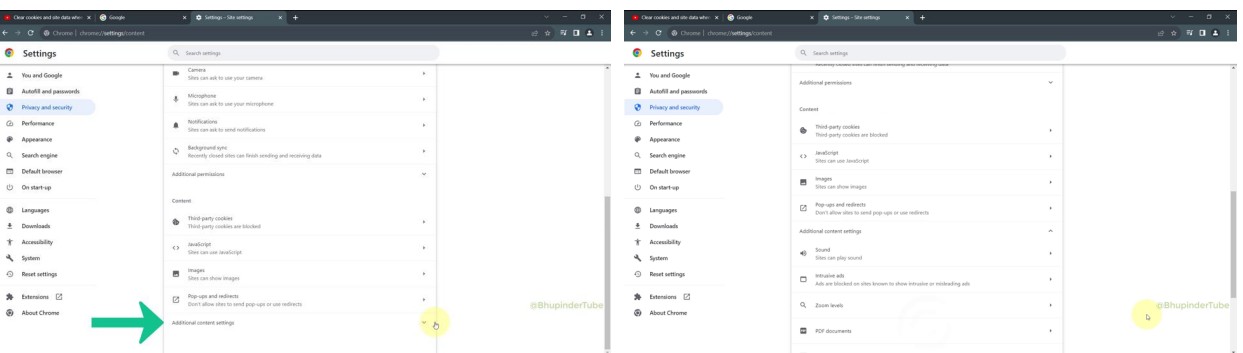

Action 3: click the [Additional content settings] button          Action 4: scroll down the settings menu

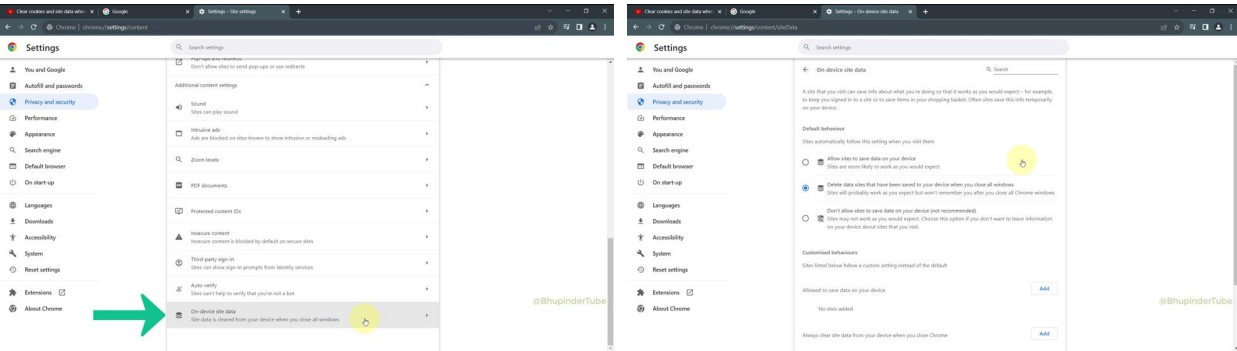

Action 5: click the [On-device site data] button          Action 6: highlight the [Delete data sites that have been saved to your device when you close all windows] option

Figure 9: An example of the final structured demonstration trajectory provided to our agent. Note that images and text are interleaved, and the text at the bottom shows the corresponding action that should be performed on the image.

**Objective**: Configure Chrome to allow specific sites to save data on the device by adding them to the allowed list.

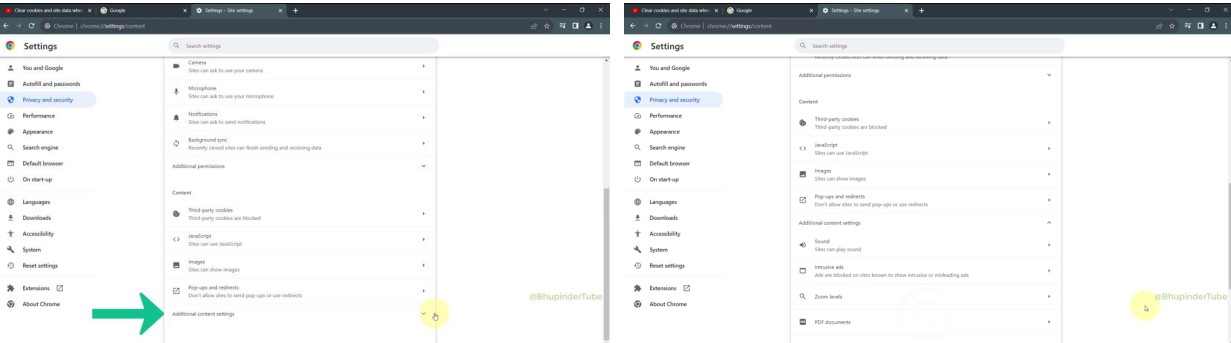

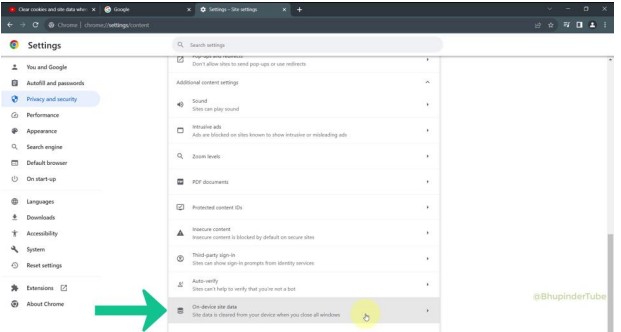

Action 1: click the [Additional content settings] button

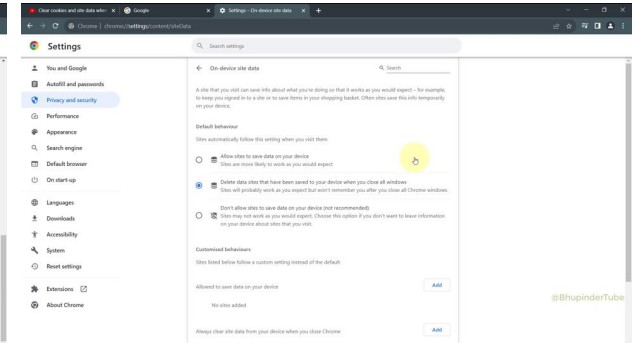

Action 2: scroll down the settings menu

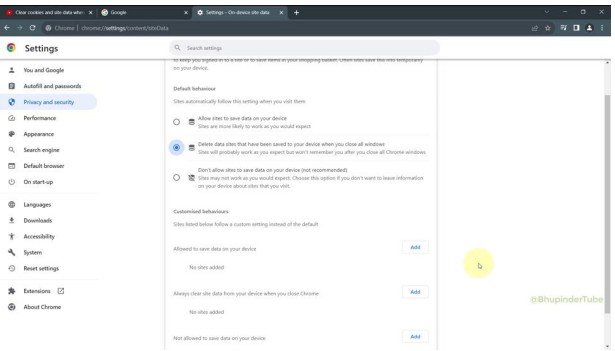

Action 3: click the [On-device site data] button

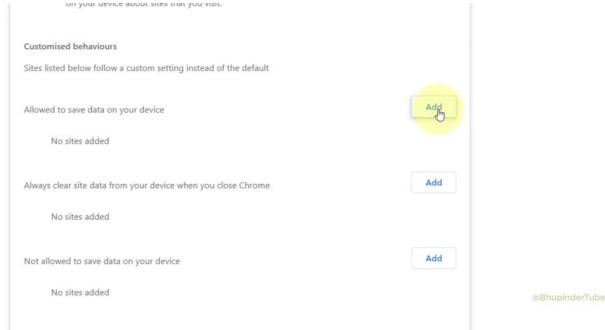

Action 4: highlight the [Delete data sites that have been saved to your device when you close all windows] option

Action 5: highlight the ['Customised behaviours'] section

Action 6: click the [Add] button next to 'Allowed to save data on your device'

Figure 10: An example of a filtered demonstration trajectory. It is filtered because it contains extra steps not required in the objective (*e.g.,* the action of "highlight the [Delete data sites that have been saved to your device when you close all windows] option"). Note that this is already a reasonable trajectory, and it is filtered only because of the unnecessary actions. For completely invalid trajectories, they will not be generated in the previous step.

Table 8: Latency of each step in our framework.

| Operation | Latency (seconds) |
|---|---|
| **Video Retrieval (per task)** | |
| Query generation | 0.1 |
| YouTube search | 1.2 |
| Video filtering step 1 | 0.6 |
| Video filtering step 2 | 19.9 |
| Total | 21.8 |
| **Video Processing (per task)** | |
| Changing frame identification | 11.2 |
| Action labeling | 260.3 |
| Action deduplication and merging | 26.6 |
| Important action filtering | 13.0 |
| Trajectory construction | 616.3 |
| Trajectory filtering | 198.7 |
| Total | 1126.1 |
| **Video application** | |
| Time per step (without videos) | 31.8 |
| Time per task (without videos) | 963.5 |
| Time per step (with videos) | 49.8 |
| Time per task (with videos) | 1434.2 |
| Additional time per step | 18.0 |
| Additional time per task | 470.7 |

```
You are a search-query generator.
Given: (1) a laptop task a user wants to accomplish, and (2) the related desktop
applications.
Goal: Produce concise web search query that a typical user would type to find a tutorial or
 guide to complete the task with those apps.

## Guidelines:
- Return only the plain query text (no labels, quotes, or operators like site:).
- Make it concise, use keywords and phrases, e.g., search for a specific application or a
specific website. Include less than 10 words in each query.
- **Abstract the general functionality or goal** behind the task. **Avoid task-specific
details** like filenames, personal data, file paths, page numbers, timestamps, etc.
- If the task involves multiple steps, focus on those that require the most application
knowledge.
- If the task involves only one app, generate exactly one query.
- If the task involves N apps (N > 1), you can generate one or more (but at most N) queries
, each at a new line. For example:
    - If there is a single primary app, generate a single query focused on that app.
    - If there are multiple primary apps, generate separate queries for each app.
```

Figure 11: Prompt used to generate search queries for video tutorials.

```
You are a helpful assistant.
Given: (1) a laptop task a user wants to accomplish, (2) the related desktop applications,
and (3) a list of retrieved videos (each with an ID, title, and description).
Goal: Select the videos that are most likely to help the user complete the task.

## Output Requirements
- Output a list of IDs for the selected videos. Output in JSON format with a single key `
selected_video_ids`. For example:

```json
{
  "selected_video_ids": [1, 5, 8, 12, 13, 17]
}
```

- Select at most 10 videos, ordered by relevance.
- If none of the videos are relevant, return an empty list.

## Guidelines:
- Exclude videos that cover unrelated applications or off-topic content, even if they are
tangentially related to the task.
```

Figure 12: Prompt used for coarse filtering of retrieved videos.

```
You are a helpful assistant.
Given: (1) a task a user wants to accomplish on their Ubuntu laptop, and (2) a retrieved
video (with title, description, transcript, and 10 sample frames).
Goal: Determine if the video (1) demonstrates the use of desktop applications on an Ubuntu
laptop, AND (2) can help the user complete the task.

## Guidelines:
- The video must contain directly applicable information that enables the user to complete
the task.
```

```
- The video must show actual usage of desktop applications (e.g., file managers, browsers,
terminals, IDEs, office software, media players, etc.) running on an Ubuntu operating
system.
- The demonstration should clearly show the interaction with the Ubuntu desktop interface (
e.g., mouse clicks, typing, application windows).
- A video should **not** be considered a demonstration if it only shows:
    - Slides or presentation decks
    - Talking-head segments (person speaking to the camera)
    - Non-desktop applications (e.g., mobile apps)

## Output Requirements
Generate your output in two sections:
OBSERVATIONS:
- List your observations about the video frames, focusing on elements that indicate whether
 desktop applications are being used on an Ubuntu laptop.
- Be specific about what you see in the frames, such as application windows, desktop
environment features, or any other relevant details.

JUDGEMENT:
- Output your judgement in JSON format with a single key `judge`. For example:
```json
{"judge": <true or false>}
```
where `true` means the video is a demonstration of using Ubuntu desktop applications and
can help the user complete the task, and `false` means otherwise.
```

Figure 13: Prompt used for content verification of retrieved videos.

```
You are given consecutive frames from a screen recording. Based on the visual change
between these frames, infer the most likely user action(s) that caused the change.

Below is the list of possible actions:
- click [button]: Click on element button. E.g., `click the [Submit] button`.
- type [text] [box]: Type text in a box. E.g., `type [Hello World] in the [search box]`.
- right click [button]: Right click a button or a link. E.g., `right click the product link
`.
- drag [element] to [position]: Drag the element to specified position. E.g., `drag [text
box] to top of the page`.
- press [key]: Press one or more keys. E.g., `press [Ctrl+F]` or `press [Esc]`.

## Instructions
1. Carefully analyze the visual changes between these frames to determine the most likely
user action(s) from the list above.
2. For each action, also indicate the start and end frame numbers (inclusive) where the
action was observed.

## Output Format
Generate your response in two sections:
OBSERVATION AND REASONING:
- Describe the visual changes step-by-step and explain what user action(s) likely caused
these changes.
ACTIONS:
- Conclude your response with all identified actions in JSON format. For example:
```json
[
    {"action": "click the [Submit] button", "start_frame": 1, "end_frame": 2},
    {"action": "type [Hello World] in the [search box]", "start_frame": 4, "end_frame": 7},
    // more actions if applicable
]
```

## Guidelines
```

```
- **Action Duration:** Include all frames from the **start of the action** (e.g., cursor
moves onto the button) to the **end of the resulting change** (e.g., page finishes loading)
.
- **Atomic Actions:** Break down actions into the smallest clear units. For example, two
consecutive button clicks should be annotated as two separate `click` actions with
different frame ranges. However, if a user types a sentence in a box, annotate it as a
single typing action.
- **Evidence Requirement:** Only annotate a click or typing action when there is **clear,
observable evidence** (e.g., a button is visibly pressed or text is entered). Ignore cursor
 movements or hovers without results.
- **Terminology Consistency:** Use the given action names. For instance, use "click"
instead of "open" or "select".
- **No Overlaps:** Frame ranges for different actions should not overlap.
- **Empty Results:** If no action can be determined, return an empty list `[]`.
```

Figure 14: Prompt used for action labeling on OSWORLD. Note that we use the same prompt on WEBARENA except for the list of possible actions.

```
You are given a list of UI actions. Your task is to detect duplicate or equivalent actions.

Two actions are considered duplicates if they have the same action type and target the same
 UI element with the same intent.
If duplicates are found, merge them into a single action that accurately represents the
combined intent.

## Output Format
Generate your response in two sections:
REASON:
- Describe the reasoning process for identifying duplicates.
MERGED ACTIONS:
- Conclude your response with all identified duplicates in JSON format. For example:
```json
[
    {"merged_action": "click the [Submit] button", "original_action_ids": [0, 4]}
    {"merged_action": "type [Hello World] in the [search box]", "original_action_ids": [2,
    3, 5]}
]
```
where `merged_action` is the merged action, and `original_action_ids` is the list of
original action IDs that were merged.

## Guidelines
- **Natural Continuation**: If two or more actions are partial segments of the same
continuous action (e.g., two typing steps that form a single sentence), merge them into one
 complete action.
- **Do Not Merge Distinct Atomic Actions**: If actions have different purposes or target
different UI elements, they must remain separate.
- **Preserve Action Type**: Keep the merged action in the same category as the original
actions.
- **Empty Results:** If no duplicates are found, return an empty list.
```

Figure 15: Prompt used for merging duplicated actions.

```
You are given (1) information of a YouTube video (title, description, and transcript) and
(2) a list of UI actions identified from the video. Your task is to select actions that are
 relevant to the task demonstrated in the video.

## Output Format
Generate your response in two sections:

ANALYSES:
- Describe the task demonstrated in the video based on the title, description, and
transcript.
- Analyze each action line by line, describe whether it is important or relevant to the
task, and explain why.

KEPT ACTIONS:
- Output your results with a JSON list containing the indices of relevant actions. For
example:

```json
[1, 2, 4, 6, 7, 8, 9, 12, 13, 17, 20]
```

## Guidelines
- **Relevance to Task**: Remove actions that are tangential or unrelated to the primary
task. Example of irrelevant actions include (but are not limited to):
  - Mouse movements or hover actions that do not lead to a click or interaction.
  - Waiting actions unless they are crucial for understanding the sequence of events.
  - Scrolling actions that do not contribute to the task.
- **Redundancy**: If multiple actions represent the same intent or outcome, keep the most
comprehensive one and remove the others.
```

Figure 16: Prompt used to filter out irrelevant actions.

```
You will be given a screenshot of a {platform} before any actions, a sequence of actions
performed on the {platform}, and a screenshot of the {platform} after the actions. Your
task is to determine whether the sequence of actions is:
1. **Coherent**: The actions are possible given the initial screenshot and lead logically
to the final screenshot.
2. **Complete**: All steps necessary to finish the task are included. If any essential step
 is missing, it is **not** complete.
3. **Contains no unnecessary actions**: Every action should be relevant to achieving the
task. Extra, unrelated steps make the sequence invalid.
4. **Matching a specific, reasonable task**: The actions clearly and fully accomplish one
identifiable goal.

If the sequence contains extra actions, is missing required steps, or does not fully
achieve the task, output "No task".
Example: If the intended task is creating a new issue, but there is no action clicking [
Create issue], this is **not** a valid task.

## Output Format
Generate your response in two sections:
OBSERVATION AND REASONING:
- Describe what is visible in the before and after screenshots.
- Go through the sequence of actions step-by-step, explaining their effects.
- State whether they accomplish a clear, reasonable task (and describe it).
- Confirm that all required steps for the task were performed – otherwise state that it is
incomplete.
- Confirm that all steps are necessary and that there are no irrelevant actions.
TASK:
- If a clear, complete task is achieved, output:
```json
{"task": "<concrete description of the task>"}
```

```
```
- If the actions are incomplete, incoherent, contain extra actions, or do not achieve any
reasonable task, output:
```json
{"task": "No task"}
```

## Example
```json
{"task": "Filter for issues labeled as bug."}
```
```

Figure 17: Prompt used to generate objectives for a sequence of actions.

```
You will be given a task description and a trajectory that consists of a screenshot of a {
platform} before any actions, a sequence of actions performed on the {platform}, and a
screenshot of the {platform} after the actions. Your job is to decide whether the
trajectory successfully accomplishes the task according to the criteria below.

## Evaluation Criteria
1. **Alignment**: The trajectory's final state clearly fulfills the task described.
2. **Coherence**: Each action logically follows from the previous observation, and no
action contradicts the task.
3. **Naturalness**: The actions resemble realistic human interaction with the {platform} in
 this context.
4. **Reasonableness**: The task is completed efficiently, without unnecessary steps,
backtracking, or overly complex methods.

## Output Format
Generate your response in two sections:
OBSERVATION AND REASONING:
- Describe what is visible in the before and after screenshots.
- Determine if the final state of the {platform} matches the expected outcome of the task.
- Go through the sequence of actions step-by-step, explaining their effects.
- Confirm that all necessary steps were performed and that each action was required.
JUDGMENT:
```json
{"judge": <true or false>, "reason": "<reasoning for the judgment>"}
```
```

Figure 18: Prompt used to filter labeled trajectories.

```
You are an autonomous intelligent agent tasked with completing a task on an Ubuntu desktop.
 Given the progress so far, the current observation, and a list of tasks with human
demonstrations, your task is to select up to 3 most relevant demonstrations that can help
you determine the next action.

Output the id of the demonstration tasks inside triple backticks, split by commas. For
example, ```2, 15, 23```.

If you think none of the demonstrations are relevant, output ```None```.
```

Figure 19: Prompt used for stage 1 selection of demonstration trajectories on OSWORLD.

```
You are an autonomous intelligent agent tasked with completing a task on an Ubuntu desktop.
 Given the progress so far, the current observation, and a list of tasks with human
demonstrations (each with a task description, the initial observation, and a sequence of
actions), your task is to select the most relevant demonstration whose detailed information
 can help you determine the next action. Assume that for each demonstration, you may later
access intermediate observations (i.e., after each action), which can further guide your
decision-making.

Output the id of the demonstration inside triple backticks. For example, ```2``` or
```3```.

If you think none of the demonstrations are relevant, output ```None```.
```

Figure 20: Prompt used for stage 2 selection of demonstration trajectories on OSWORLD.

```
You are an autonomous intelligent agent tasked with navigating a web browser.

Given the progress so far, the current observation, and a list of demonstration tasks, your
 task is to select up to 3 most relevant demonstrations that can help determine the next
action.

Output your response in three steps.
1. First, analyze whether you already have enough knowledge to determine the next action.
2. If not, inspect the demonstrations to see if they contain the required information.
3. Output your final decision in the following two formats:

Case 1 – no demonstration needed: If you already have enough knowledge to determine the
next action, or none of the demonstrations are relevant, output None inside triple
backticks. For example, ```None```.
Case 2 – demonstration(s) needed: If you need the detailed information from some
demonstrations, output the id of the demonstration tasks inside triple backticks, split by
commas. For example, ```2, 15```.
```

Figure 21: Prompt used for stage 1 selection of demonstration trajectories on WEBARENA.

```
You are an autonomous intelligent agent tasked with navigating a web browser.

Given the progress so far, the current observation, and a list of demonstration tasks
extracted from a video (each with a task description, the initial observation, and a
sequence of actions), your task is to select the most relevant demonstration that can help
determine the next action.

Output your response in three steps.
1. First, analyze whether you already have enough knowledge to determine the next action.
2. If not, inspect the demonstrations to see if they contain the required information.
3. Output your final decision in the following two formats:

Case 1 – no demonstration needed: If you already have enough knowledge to determine the
next action, or none of the demonstrations are relevant, output None inside triple
backticks. For example, ```None```.
Case 2 – demonstration needed: If you need the detailed information from a specific
demonstration, output the id of the demonstration inside triple backticks. For example,
```2``` or ```3```.
```

Figure 22: Prompt used for stage 2 selection of demonstration trajectories on WEBARENA.

```
You are an autonomous intelligent agent tasked with completing a task on an Ubuntu desktop.
 You have previously found a demo trajectory for a related task, and now your task is to
determine if the demo is still relevant and useful for choosing your next action.

You will be given:
- The current state of your task and environment (i.e., your progress and current
observation)
- The demo trajectory you found previously

Respond with one of the following, inside triple backticks:
    * ```Yes``` - if the demo trajectory is still relevant and can help you determine the
    next action.
    * ```No``` - if the demo trajectory is no longer relevant or helpful.
```

Figure 23: Prompt used to determine whether to preserve the previously selected trajectory.

