# OpenReview forum: "Learning from Online Videos at Inference Time for Computer-Use Agents"
_TMLR — Accepted by TMLR_

### Review · Reviewer_n2V8 · 2025-11-12

**Summary Of Contributions:**

The paper presents a framework to enable computer-use agents to learn from online video tutorials at inference time. The core idea is to mimic how humans learn: by searching for relevant tutorials, watching short, relevant segments, and imitating the demonstrated actions. The proposed pipeline consists of three main stages: (1) **Video Retrieval**, which finds and filters relevant tutorial videos from online sources like YouTube; (2) **Video Processing**, which uses a Vision-Language Model (VLM) to convert raw videos into structured, actionable trajectories, each containing a sequence of actions, observations, and a high-level textual objective; and (3) **Video Application**, which at each step of a task dynamically selects the single most relevant trajectory to provide as in-context guidance to the agent. The method is evaluated on two standard benchmarks, OSWORLD and WEBARENA, where it shows improvements over baseline agents and variants that use only textual information.

**Strengths:**
*   The paper addresses the critical limitation of domain-specific knowledge in current computer agents with a well-motivated and intuitive solution inspired by human learning. The focus on inference-time learning from videos is a practical and timely direction for the field.
*   The framework demonstrates good empirical results, consistently outperforming state-of-the-art base agents (Jedi, AgentOccam) and other guidance methods on two diverse and widely-used benchmarks. The performance gains in some challenging domains are notable.
*   The ablation studies convincingly demonstrate the importance of several key components of the proposed framework, highlighting the value of dynamic trajectory selection, action filtering, and the visual information present in videos.

**Weaknesses:**
*   The proposed pipeline appears to be an engineering-heavy workflow that is computationally expensive and likely suffers from high latency. The video processing stage seems to scale quadratically with the number of actions, and the inference-time selection mechanism involves VLM calls at every step. The paper completely omits any discussion or measurement of these practical costs, which undermines its claim as an effective *inference-time* method.
*   The paper lacks qualitative analysis and a rigorous examination of failure cases. The aggregated success rates obscure the specific reasons for failure, making it difficult to understand the bottlenecks in the pipeline (e.g., poor video retrieval, incorrect action labeling, or flawed trajectory selection).
*   The methodology lacks sufficient detail for reproducibility. Key steps, such as the generation and filtering of trajectories from "all subsequences of actions" (Algorithm 1), are described too abstractly. The $O(N^2)$ complexity of this step is a significant scalability concern that is not addressed.

**Audience:**

Yes

**Audience Explanation:**

The field of autonomous agents is a highly active area of research. The problem of endowing agents with procedural knowledge to operate diverse software is a fundamental challenge. Researchers and practitioners in machine learning, particularly those focused on agents, reinforcement learning, and multimodal models, would be very interested in this paper's approach to inference-time learning from unstructured videos. The topic is timely and the empirical findings, despite their current shortcomings in presentation, point to a valuable research direction.

**Broader Impact Concerns:**

The proposed agent could be misused if it learns from tutorials for malicious activities. Furthermore, its reliance on unvetted online content raises safety and reliability concerns, as it may imitate incorrect or unsafe procedures.

**Claims And Evidence:**

No

**Claims Explanation:**

While the paper presents a promising direction, the evidence provided is not yet sufficient to fully support its claims in a convincing manner. First, the claim of practicality for an "inference-time" method is unsubstantiated, as the paper provides no analysis of the significant computational cost and latency its pipeline likely introduces. Second, the evidence for effectiveness is incomplete; the paper fails to investigate or explain counter-intuitive results, such as why adding video transcripts lowers performance on several WEBARENA tasks, and provides no failure analysis to add depth to the aggregate scores. Finally, the evidence for the method itself is weakened by a high-level methodological description that omits key implementation details, making it difficult to assess the correctness and robustness of the experiments.

**Requested Changes:**

1.  The paper must include a dedicated analysis of the computational costs. This should include: (a) the complexity of the video processing pipeline; (b) the measured latency added at each inference step by the trajectory selection mechanism; and (c) a discussion of the practicality of the approach and potential optimizations. This is essential to validate the "inference-time" claim.
2.  The authors must supplement the quantitative results with a qualitative analysis. This should include: (a) an analysis of several failure cases, identifying the root cause of the failure (e.g., retrieval, processing, or selection error); and (b) an explanation for counter-intuitive results, specifically the performance degradation observed when using only transcripts on WEBARENA (Table 2).
3.  The methodology section requires more detail to ensure reproducibility. Specifically, explain the process for "Demonstration trajectory construction," addressing its $O(N^2)$ complexity and scalability. Also, clarify how the "3.6 videos per task" statistic was calculated.
4.  The paper would be significantly improved by an appendix section with a concrete, step-by-step example of the pipeline. Show key frames from a video, the extracted actions, the generated objective, and the final structured trajectory provided to the agent.

---

> ### Author Response · Authors · 2025-12-11
>
> We thank the reviewer for the constructive feedback. We address each concern below and have revised the paper accordingly.
>
> **1. The paper should include a dedicated analysis of the computational costs.**
>
> Thanks for the suggestion. We have conducted a detailed analysis of the computational costs of our method. Specifically, we measure the additional costs for the three stages of our method, as illustrated in Figure 1: (1) video retrieval and filtering, (2) video processing, and (3) video application. The following table summarizes the additional latency for each step, where the time is measured on a random subset of 100 tasks on OSWorld. For a fair comparison, we run our method and the baseline (Jedi) with the same configuration during the same time period (to control for OpenAI API latency variations). For video retrieval and processing, we serve the LLM/VLM with vLLM on 8 AMD MI250 GPUs.
> | Operation  | Latency (seconds) |
> |---|---|
> | **Video Retrieval (per task)** |  |
> | Query generation | 0.1 |
> | YouTube search | 1.2 |
> | Video filtering step 1 | 0.6 |
> | Video filtering step 2 | 19.9 |
> | Total  | 21.8 |
> | **Video Processing (per task)** |  |
> | Changing frame identification | 11.2 |
> | Action labeling | 260.3 |
> | Action deduplication and merging | 26.6 |
> | Important action filtering | 13.0 |
> | Trajectory construction | 616.3 |
> | Trajectory filtering | 198.7 |
> | Total  | 1126.1 |
> | **Video application** |  |
> | Time per step (without videos) | 31.8 |
> | Time per task (without videos) | 963.5 |
> | Time per step (with videos) | 49.8 |
> | Time per task (with videos) | 1434.2 |
> | Additional time per step | 18.0 |
> | Additional time per task | 470.7 |
>
> As can be observed, among the three stages, video retrieval has minimal latency because it involves relatively lightweight operations. For video processing, most of the time is spent on trajectory construction, since it needs to iterate over subsequences of actions to generate objectives. An implementation detail that we previously omitted in the paper is that we limit each trajectory to contain at most 15 actions. This constraint reduces the time complexity for trajectory construction from $O(N^2)$ to $O(N)$. We have modified Algorithm 1 in the paper to reflect this detail.
>
> Crucially, **the first two stages can be done offline before the actual task inference**, which reduces the actual latency of the user query. To achieve this, we can follow [1] to synthesize a set of seed task queries and use them to retrieve, filter, and process videos before seeing the real user queries. When a real user query arrives, we can retrieve similar synthetic queries and use the corresponding already-processed videos for inference.
>
> For the online part, although the trajectory selection incurs around 48.9% additional costs per task compared to the baseline that doesn’t have video information, we argue that this cost is acceptable considering the performance improvements of our method. Overall, we believe our method is **well-suited as an inference-time approach** because (1) it can be applied to individual user queries without large-scale data collection; (2) it does not require parameter updates; and (3) the additional cost is minimal compared to large-scale training (e.g., [2] reports that doing supervised training requires roughly 2944 GPU hours on H100).
>
> Finally, we also discuss several practical optimizations to further reduce computational costs. First, the first two stages can be moved offline as described above. Second, the retrieved videos can be processed at a lower frame rate (currently 2 frames per second) when identifying changing frames and labeling actions, so that both the number of actions and constructed trajectories will be reduced. Third, at each inference step of the final stage, we can ask the agent to first self-evaluate its own knowledge, and if the agent already has enough information to decide the next action, we can skip the trajectory selection and directly generate the next action without videos. In summary, many implementation variations can be introduced to further reduce the overhead, making the method more practical as an inference-time solution.
>
> [1] Zhang et al., TongUI: Building Generalized GUI Agents by Learning from Multimodal Web Tutorials.
>
> [2] Lu et al., VideoAgentTrek: Computer Use Pretraining from Unlabeled Videos.

---

> > ### Author Response · Authors · 2025-12-11
> >
> > **2. The authors should add qualitative analysis, including analysis of failure cases and counterintuitive results.**
> >
> > We have conducted an error analysis for our method. Particularly, we manually examine 20 tasks in OSWorld where the baseline agent is correct, but our method is wrong. We categorize the error causes into two groups: one group that is related to our video pipeline and the other that is not. We have added **Appendix B** in the paper to discuss these failure cases and show concrete examples.
> >
> > For the errors that can be attributed to our video pipeline (6/20), we identify the following reasons:
> > - Mismatch between the video’s UI version and the agent’s UI version (1/20). We identify a case where the video shows how to click “Display Google Chrome in this language” in order to change the default language, but the agent interacts with a different Chrome version lacking this button, leading to confusion and repeated clicks.
> > - The agent already completed the task, but followed the video to do unnecessary extra steps (2/20). For example, for a task that requires searching for specific flights. The agent applied the required constraints and obtained the search results, but it clicked on one of the flights for subsequent purchase.
> > - The agent didn’t select the useful trajectory or didn’t find useful videos (2/20). For a task that requires showing the power percentage, the pipeline successfully constructs a trajectory for showing the power percentage in settings, but this trajectory is not selected during inference. In another task that requires exporting the address book from Thunderbird and converting it to an xlsx file, the system fails to find a video demonstrating exporting in Thunderbird, retrieving only a video about LibreOffice file conversion. In both cases, the agent lacks useful guidance and fails the task.
> > - The agent didn’t follow the selected trajectory (1/20). For a task requiring copying and pasting a transposed table, the agent selected a valid trajectory but didn’t follow the trajectory to click the buttons and instead used a hotkey, which led to a wrong outcome.
> >
> > For the errors **not related** to our video pipeline (14/20), we identify three main groups:
> > - Grounding issues (6/20). In these cases, the agent planned the correct next action (e.g., scroll down the list to find a specific item or drag an element to a specific position). However, to execute these actions, it needs to generate code that specifies the precise amount for scrolling and the precise (x, y) coordinates for dragging and clicking. When the coordinates are wrong, the actions will not be successfully executed, and the agent keeps repeating the same actions. This is also noted as prominent in OSWorld [3].
> > - Other issues (6/20). These failures are caused by unrelated system errors, such as network connection errors, or the agent is distracted by pop-up windows while browsing the web.
> > - Evaluation errors (2/20). In two tasks, the agent correctly completed the task (e.g., flip an image horizontally), but the evaluation marked it as incorrect.
> >
> > Overall, we observe that **a majority of the failure cases are not caused by adding video information**. Among video-related failures, the main issues are (1) **video retrieval**, where the system couldn’t find useful videos; and (2) **video application**, where the agent couldn’t adapt when the UI version in the video is different from what it sees, or it follows the videos to do extra actions that are not required in the tasks. This suggests that future improvements can be made by improving the agent’s ability to adapt the selected trajectory to the testing task.
> >
> > Regarding the counter-intuitive performance drop with transcripts on WebArena, especially on the Reddit domain, we manually inspect the tasks where the baseline agent (AgentOccam) completes correctly, but the agent with transcript information fails. We find two categories of failure cases:
> > - Mismatch between transcripts and the agent’s action space. We observe that for 5 tasks that require liking/disliking all posts of a particular user, the transcript specifically mentions how to visit a user’s page by directly typing the URL in the browser. However, for AgentOccam, the agent’s action space is limited to simple clicking and typing, and it cannot jump to a specific URL. This discrepancy misleads the agent and hurts performance.
> > - Other issues. We also identify other errors that are not directly related to the transcript information. For example, the agent created posts in the wrong subreddit, or it was distracted by the branching operation and didn’t complete the required task.
> >
> > In general, we find that **text-only transcripts often contain ambiguous information**, e.g., “click this and put that here”. Without the complementary visual information, such guidelines could distract the agent and hurt performance.
> >
> > [3] Xie et al., OSWORLD: Benchmarking Multimodal Agents for Open-Ended Tasks in Real Computer Environments.

---

> > > ### Author Response · Authors · 2025-12-11
> > >
> > > **3. The methodology section requires more details, e.g., explain the process for demonstration trajectory construction and clarify how the number of 3.6 videos per task is calculated.**
> > >
> > > We have updated Algorithm 1 to reflect the actual implementation we used to construct demonstration trajectories. Specifically, we iterate over all subsequences of actions that have a length shorter than or equal to 15. For each subsequence of actions, we provide the VLM with all actions (in text), as well as the beginning and end frames of the subsequence (we omit the intermediate frames for efficiency). The VLM is instructed with the prompt in Figure 16 to generate an objective for this subsequence, or None if the actions do not achieve any reasonable task. We then filter the labeled trajectories using the prompt in Figure 17 to make sure the final trajectories are coherent and complete. Because we limit each trajectory to at most 15 actions, the complexity of this step reduces from $O(N^2)$ to $O(N)$. Additionally, since we only provide the VLM with start and end frames without intermediate frames, the latency for each VLM query is relatively small, making it scalable to videos with hundreds of actions.
> > >
> > > The number of 3.6 videos per task is calculated on the subset of 211 tasks in OSWorld, on which our system finds at least one relevant video. These are the videos after our filtering process, i.e., after removing videos that do not demonstrate the use of desktop applications on an Ubuntu laptop, or those that are irrelevant to the task.
> > >
> > > We have updated our paper to include these details.
> > >
> > > **4. The paper would be improved by adding a concrete, step-by-step example of the pipeline.**
> > >
> > > Thanks for the suggestion. We have added **Appendix C** in our paper, which includes an example of our video processing pipeline. The added example shows how our pipeline identifies changing frames, labels actions, and generates an objective that matches the task requirement. Please refer to the updated PDF for more details.

---

### Review · Reviewer_iEPs · 2025-11-20

**Summary Of Contributions:**

The paper proposes a two-stage pipeline consisting of **video retrieval** followed by **video processing**. Third stage is application so i am skipping in the summary.

In the **video retrieval** stage, an agent is tasked with finding videos relevant to a given instruction or task. First, the task for the agent is defined. The agent then retrieves the top 50 candidate videos using standard search signals (e.g., title and basic metadata), without any sophisticated reasoning at this point. Next, a large language model (LLM) is used to filter these results based on the actual video content, narrowing them down to the top 10 candidates. For each of these 10 videos, the system samples 10 frames and feeds them into a vision–language model (VLM) to assess whether the video is likely to be relevant to the target task. **The exact sampling strategy (e.g., whether this corresponds to 10 frames total, 10 fps, or another rate)** is not fully clear and is left somewhat underspecified in the paper. After this step, the system has a set of videos that are contextually appropriate and exhibit the right kinds of actions, but they are still only *potential* matches for the full task trajectory.

The second stage is **video processing**. Here, the system resamples the selected videos at 2 frames per second and performs frame differencing. Large differences between frames are interpreted as potential boundaries between different subtasks or trajectories. Each such trajectory segment is then passed to a VLM, which assigns a label describing the underlying action or subtask. This produces a sequence of subtasks. For example, in a “cooking” video, the system might identify subtasks such as “cutting onions,” “cutting tomatoes,” and “mixing ingredients,” even though the video might only be labeled at a high level as “cooking.”

However, identifying subtasks alone does not guarantee that the full, desired task is demonstrated correctly. To address this, the authors introduce a **relabeling and verification** step. The VLM-generated trajectory labels (e.g., “cutting onions,” “cutting tomatoes,” “mixing together”) are passed to an LLM, which checks whether these subtasks indeed form part of the intended overall task. This process is iterative, allowing the agent to gradually refine its understanding of which videos truly show how to complete the target task from start to finish.

The authors evaluate their method on two widely used benchmarks and report incremental (rather than dramatic) performance improvements. Nonetheless, the combination of video retrieval and structured video processing is presented as a novel and interesting approach. The paper also provides detailed implementation information and extensive references to prior work, which appear sound and well-chosen. Overall, it offers a clear, methodical framework for teaching an agent to learn tasks from videos.

** I am aware of author mentioning uniform sampling, this raises more question such as what if the video is 2-3 min long or very short. Both can have little information to provide due sparsity or due to repetative frames.

**Additional Comments:**

I am aware of author mentioning uniform sampling, this raises more question such as what if the video is 2-3 min long or very short. Both can have little information to provide due sparsity or due to repetative frames.
Also can you add few more visualization of the approach. The one you have provided is good, but the flow can be more cleaned and well presented.

**Audience:**

Yes

**Audience Explanation:**

This paper is not research or innovative in terms of new design, task or new discovery rather its improvement of agent task. Agent task is new domain, so this paper definietly contributes to innovation.

**Claims And Evidence:**

Yes

**Claims Explanation:**

Yes, the paper has been benchmarked and shows clear evidence. Also the paper provides implementation details in supplementary section.

**Requested Changes:**

Concerns around sampling rate used while video retrieval

---

> ### Author Response · Authors · 2025-12-13
>
> We thank the reviewer for the constructive feedback. We address each concern below and have revised the paper accordingly.
>
> **1. Concerns around sampling rate used in video retrieval.**
>
> We would like to clarify that the uniformly sampled 10 frames are only used in the video filtering stage, not for action labeling or trajectory construction. While this sampling is sparse, it is **sufficient for the purpose of relevance filtering**. Concretely, we provide the VLM with the video title, description, transcript, and the 10 uniformly sampled frames, and the VLM determines if the video contains relevant information to the user query and demonstrates the usage of Ubuntu desktop applications. The description and transcript already carry most of the semantic information needed to determine the relevance. The sampled frames mainly serve as a visual sanity check to verify that the video indeed demonstrates Ubuntu desktop application usage. For example, the frames allow the VLM to filter out videos that primarily show slide presentations, talking-head explanations, or mobile phone interfaces, even if the transcript appears superficially relevant.
>
> Because this stage only requires a high-level skim of the video rather than a fine-grained understanding, sparse uniform sampling is usually sufficient. Additionally, we limit the retrieved videos to be at least 30 seconds long, so the sampled frames are unlikely to be dominated by repetitive or non-informative content.
>
> **2. Provide more visualization of the approach.**
>
> We have added **Appendix C** in our paper, which includes an example walking through our video processing pipeline. The added example shows how our pipeline identifies changing frames, labels actions, generates an objective that matches the task requirement, and filters invalid trajectories. Please refer to the updated PDF for more details.

---

### Review · Reviewer_JfdE · 2025-11-23

**Summary Of Contributions:**

### **Summary:**
This paper proposes an inference-time framework that enables computer-use agents to learn from online tutorial videos to improve task performance on desktops and the web, without any additional parameter updates. The method retrieves relevant videos, converts them into structured demonstration trajectories (with inferred actions, screenshots, and textual objectives), and then dynamically selects a single most relevant trajectory at each step to provide in-context guidance to the agent.

### **Key technical components:**
1. Video retrieval: LLM-generated search queries, YouTube search, and LLM/VLM-based coarse and content filtering.

2. Video processing:
    (i) VLM-based action labeling across frame transitions.
    (ii) Filtering of irrelevant actions (e.g., mouse hovering, tangential interactions).
    (iii) Generation of sub-trajectories with VLM-synthesized natural language objectives.
    (iv) Validation/filtering of trajectories for coherence and completeness.

3. Video application at inference time:
    (i) A two-stage trajectory selection mechanism (objective-based coarse ranking + detailed inspection using initial screen and action sequence).
    (ii) A context-preservation heuristic that decides whether to continue using the previous trajectory or re-select a new one.

4. Experiments on OSWorld-Verified and WebArena show that the method:
    (i) Outperforms strong base agents Jedi and AgentOccam.
    (ii) Outperforms variants that use only textual tutorials or only transcripts, indicating that structured, visually grounded trajectories are more useful than unstructured text.
    (iii) Benefits from more relevant videos and from design choices such as action filtering and trajectory segmentation.

### **Strengths:**
1. Clear, compelling motivation: agents lack domain-specific procedural knowledge; humans often rely on short tutorial segments to fill this gap.
2. Well-structured pipeline that is modular and compatible with existing agents.
3. Evaluated on two widely used, realistic benchmarks (OSWorld, WebArena) with strong baselines.

### **Ablations convincingly show the importance of:**
1. Trajectory segmentation and dynamic selection.
2. Filtering irrelevant actions.
3. Incorporating visual information.
4. Training-free at test time; relatively easy to plug into other frameworks.

### **Weaknesses:**
1. Heavy reliance on VLM quality for accurate action labeling and trajectory construction; limited analysis of compounding errors.
2. Limited qualitative analysis of failure modes (e.g., incorrect trajectories, UI mismatches).
3. Retrieval stage effectiveness is not quantitatively evaluated (no precision/recall of video filtering).
4. No systematic measurement of latency and compute overhead, which are crucial for real deployments.
5. Dependence on external video platforms (YouTube) raises questions about temporal reproducibility and data availability.

**Additional Comments:**

1. The paper is clear and well-organized, with a logical progression from motivation to method, experiments, and analysis.
2. The human analogy (“search, skim, imitate short segments that match subgoals”) is not just rhetorical; it is operationalized in the pipeline via segmentation and dynamic selection.
3. The two-stage selection framework is conceptually clean and appears to be a major practical contributor; further analysis of selection stability across long trajectories (e.g., how often trajectories are switched) would be interesting.
4. Including confidence intervals or task-level variance for success rates would improve the statistical rigor of the reported results.
5. The work is nicely positioned as complementary to training-time video pretraining efforts; a short explicit subsection comparing pros/cons (e.g., compute, flexibility, data requirements) of inference-time vs training-time usage would be a useful addition.

**Audience:**

Yes

**Audience Explanation:**

This work sits at the intersection of:
1. Computer-use / GUI agents
2. Multimodal LLMs/VLMs
3. Imitation learning from observations (video)
4. Test-time augmentation / retrieval-based methods

These are active and fast-moving research areas. The specific idea of using online tutorial videos at inference time, without retraining, to guide agents is:
1. Novel in its test-time focus (complementing existing training-time video-pretraining works).
2. Immediately relevant to practitioners building real-world agents (desktop or web).
3. Conceptually interesting for the broader community interested in in-context learning and multimodal grounding.

Given the prominence of OSWorld/WebArena and the increasing interest in practical agent systems, the paper’s findings are clearly of interest to a non-trivial portion of TMLR’s readership.

**Broader Impact Concerns:**

The paper should expand the Broader Impact discussion to address:

### **Copyright and Terms of Service:**
1. The system systematically retrieves and processes YouTube tutorial videos. Clarify:
    (i) Compliance with YouTube’s terms of service.
    (ii) Whether only metadata + transcripts + frame samples are used, and how they are stored.
    (iii) Any restrictions on sharing or redistributing processed trajectories derived from copyrighted videos.

### **Security and Safety Risks:**
1. Tutorial videos may instruct sensitive or dangerous operations (e.g., editing system settings, deleting data). Agents blindly following them may perform harmful actions.
2. Discuss mitigations such as:
    (i) Restricting actions to safe environments (sandboxed VMs).
    (ii) Detecting and filtering risky operations through additional safety filters.

### **Bias and Representational Issues:**
1. If the video corpus is skewed toward certain software versions, platforms, or user workflows, the agent might inherit these biases.
2. Consider how this affects generalization and fairness across different user populations and setups.

### **Privacy:**
1. Clarify that only publicly available video content is used and that the method does not access or rely on private account-linked resources.

**Claims And Evidence:**

Yes

**Claims Explanation:**

## The main empirical claims are:
1. Inference-time use of video-derived trajectories improves agent performance on computer-use benchmarks.
2. Structured, visually grounded trajectories are more helpful than text-only tutorials or transcripts.
3. The proposed design choices (segmentation, selection, filtering, visual inclusion) are each important contributors.

## These are supported as follows:
1. Quantitative results:
    (i) On OSWorld, the proposed method improves success rate over Jedi by +3.5 on tasks with available videos and +2.1 overall (Table 1).
    (ii) On WebArena, the method improves over AgentOccam and a transcript-only variant, with particularly large gains (+10.2) on the more challenging GitLab domain (Table 2).

2. Ablation studies:
    (i) Removing trajectory selection and always providing the full video trajectory significantly hurts performance, showing the necessity of segmenting videos and dynamically selecting relevant trajectories.
    (ii) Removing action filtering degrades performance, demonstrating the value of removing irrelevant UI actions.
    (iii) Removing visual information from trajectories (using text-only) reduces performance, supporting the claim that visual context provides additional, non-redundant information (Table 4).

3. Scale analysis:
    (i) The paper shows that having more videos per task (e.g., 3.6 vs 1) improves success rate (Table 3), consistent with the claim that abundant online videos can be leveraged at inference time.
    (ii) While some implementation details (e.g., error analysis of the VLM labeling, robustness under noisy or misleading videos) are underexplored, the core quantitative claims are well-supported by standard benchmarks and strong baselines.

**Requested Changes:**

Below are proposed adjustments. Each item is marked as Critical (necessary for a positive recommendation) or Recommended (would strengthen the paper).

### **Critical:**
1. Quantify computational cost and latency:
    (i) Report end-to-end runtime per task and per step with and without video integration.
    (ii) Break down additional overhead from:
        (a) Video retrieval and processing (offline vs. online).
        (b) Per-step VLM/LLM calls for trajectory selection.
    (iii) Clarify whether the pipeline is practically deployable under typical inference budgets.

2. Provide failure mode and qualitative error analysis:
    (i) Show concrete examples where:
        (a) The selected demonstration trajectory is clearly mismatched with the agent’s current state.
        (b) The VLM mislabels actions or produces incoherent objectives.
        (c) Visual mismatches (OS differences, app versions, window layouts) lead to incorrect guidance.
    (ii) Summarize the dominant failure patterns and how often they occur.

3. Discuss robustness to noisy or misleading videos:
    (i) Analyze or discuss how the system behaves if:
        (a) Tutorials are outdated (UI version changed).
        (b) Videos contain mixed or irrelevant workflows.
        (c) There is adversarial or low-quality content.
    (ii) If possible, provide an experiment or at least a controlled case study illustrating robustness or failure.

### **Recommended:**
4. Quantify video retrieval effectiveness:
    (i) Provide statistics on:
        (a) Fraction of initially retrieved videos that pass coarse and content filtering.
        (b) Human-judged relevance of filtered videos for a small subset.
    (ii) This would better justify the retrieval stage and illustrate how often the pipeline can find useful tutorials.

5. Add qualitative trajectory visualizations:
    (i) Include 1–2 figures showing:
        (a) Example video frames, labeled actions, and synthesized objective.
        (b) A “good” trajectory vs a “filtered-out” or invalid trajectory.
    (ii) This will improve the interpretability of the VLM-driven processing steps.

6. Clarify scaling behavior and limits:
    (i) Explore or at least discuss:
        (a) Impact of increasing the number of videos per task beyond the current average.
        (b) Potential issues with very large demonstration pools (e.g., context length, selection noise).

7. Improve reproducibility of video-based experiments:
    (i) Since YouTube search is non-deterministic over time, consider releasing:
        (a) The list of video IDs/URLs used.
        (b) Pre-processed trajectories (or a subset) as a dataset.
    (ii) This is important for replicability and future comparison.

8. Explicate assumptions on VLM capabilities:
    (i) Discuss what level of VLM performance is required for the method to be effective (e.g., approximate accuracy on action detection, objective coherence).
    (ii) This will help readers assess applicability under different model choices.

---

> ### Author Response · Authors · 2025-12-13
>
> We thank the reviewer for the constructive feedback. We address each concern below and have revised the paper accordingly.
>
> **1. Quantify computational cost and latency.**
>
> Thanks for the suggestion. We have conducted a detailed analysis of the computational costs of our method. Specifically, we measure the additional costs for the three stages of our method, as illustrated in Figure 1: (1) video retrieval and filtering, (2) video processing, and (3) video application. The following table summarizes the additional latency for each step, where the time is measured on a random subset of 100 tasks on OSWorld. For a fair comparison, we run our method and the baseline (Jedi) with the same configuration during the same time period (to control for OpenAI API latency variations). For video retrieval and processing, we serve the LLM/VLM with vLLM on 8 AMD MI250 GPUs.
> | Operation  | Latency (seconds) |
> |---|---|
> | **Video Retrieval (per task)** |  |
> | Query generation | 0.1 |
> | YouTube search | 1.2 |
> | Video filtering step 1 | 0.6 |
> | Video filtering step 2 | 19.9 |
> | Total  | 21.8 |
> | **Video Processing (per task)** |  |
> | Changing frame identification | 11.2 |
> | Action labeling | 260.3 |
> | Action deduplication and merging | 26.6 |
> | Important action filtering | 13.0 |
> | Trajectory construction | 616.3 |
> | Trajectory filtering | 198.7 |
> | Total  | 1126.1 |
> | **Video application** |  |
> | Time per step (without videos) | 31.8 |
> | Time per task (without videos) | 963.5 |
> | Time per step (with videos) | 49.8 |
> | Time per task (with videos) | 1434.2 |
> | Additional time per step | 18.0 |
> | Additional time per task | 470.7 |
>
> As can be observed, among the three stages, video retrieval has minimal latency because it involves relatively lightweight operations. For video processing, most of the time is spent on trajectory construction, since it needs to iterate over subsequences of actions to generate objectives. An implementation detail that we previously omitted in the paper is that we limit each trajectory to contain at most 15 actions. This constraint reduces the time complexity for trajectory construction from $O(N^2)$ to $O(N)$. We have modified Algorithm 1 in the paper to reflect this detail.
>
> Crucially, **the first two stages can be done offline before the actual task inference**, which reduces the actual latency of the user query. To achieve this, we can follow [1] to synthesize a set of seed task queries and use them to retrieve, filter, and process videos before seeing the real user queries. When a real user query arrives, we can retrieve similar synthetic queries and use the corresponding already-processed videos for inference.
>
> For the online part, although the trajectory selection incurs around 48.9% additional costs per task compared to the baseline that doesn’t have video information, we argue that this cost is acceptable considering the performance improvements of our method.
>
> Overall, we believe our method is practically deployable at inference time. Additionally, several implementation variations can be introduced to further reduce the overhead. First, the first two stages can be moved offline as described above. Second, the retrieved videos can be processed at a lower frame rate (currently 2 frames per second) when identifying changing frames and labeling actions, so that both the number of actions and constructed trajectories will be reduced. Third, at each inference step of the final stage, we can ask the agent to first self-evaluate its own knowledge, and if the agent already has enough information to decide the next action, we can skip the trajectory selection and directly generate the next action without videos.
>
>
> [1] Zhang et al., TongUI: Building Generalized GUI Agents by Learning from Multimodal Web Tutorials.

---

> ### Author Response · Authors · 2025-12-13
>
> **2. Provide failure mode and qualitative analysis.**
>
> We have conducted an error analysis for our method. Particularly, we manually examine 20 tasks in OSWorld where the baseline agent is correct, but our method is wrong. We categorize the error causes into two groups: one group that is related to our video pipeline and the other that is not. We have added **Appendix B** in the paper to discuss these failure cases and show concrete examples.
>
> For the errors that can be attributed to our video pipeline (6/20), we identify the following reasons:
> - Mismatch between the video’s UI version and the agent’s UI version (1/20). We identify a case where the video shows how to click “Display Google Chrome in this language” in order to change the default language, but the agent interacts with a different Chrome version lacking this button, leading to confusion and repeated clicks.
> - The agent already completed the task, but followed the video to do unnecessary extra steps (2/20). For example, for a task that requires searching for specific flights. The agent applied the required constraints and obtained the search results, but it clicked on one of the flights for subsequent purchase.
> - The agent didn’t select the useful trajectory or didn’t find useful videos (2/20). For a task that requires showing the power percentage, the pipeline successfully constructs a trajectory for showing the power percentage in settings, but this trajectory is not selected during inference. In another task that requires exporting the address book from Thunderbird and converting it to an xlsx file, the system fails to find a video demonstrating exporting in Thunderbird, retrieving only a video about LibreOffice file conversion. In both cases, the agent lacks useful guidance and fails the task.
> - The agent didn’t follow the selected trajectory (1/20). For a task requiring copying and pasting a transposed table, the agent selected a valid trajectory but didn’t follow the trajectory to click the buttons and instead used a hotkey, which led to a wrong outcome.
>
> For the errors **not related** to our video pipeline (14/20), we identify three main groups:
> - Grounding issues (6/20). In these cases, the agent planned the correct next action (e.g., scroll down the list to find a specific item or drag an element to a specific position). However, to execute these actions, it needs to generate code that specifies the precise amount for scrolling and the precise (x, y) coordinates for dragging and clicking. When the coordinates are wrong, the actions will not be successfully executed, and the agent keeps repeating the same actions. This is also noted as prominent in OSWorld [2].
> - Other issues (6/20). These failures are caused by unrelated system errors, such as network connection errors, or the agent is distracted by pop-up windows while browsing the web.
> - Evaluation errors (2/20). In two tasks, the agent correctly completed the task (e.g., flip an image horizontally), but the evaluation marked it as incorrect.
>
> Overall, we observe that **a majority of the failure cases are not caused by adding video information**. Among video-related failures, the main issues are (1) **video retrieval**, where the system couldn’t find useful videos; and (2) **video application**, where the agent couldn’t adapt when the UI version in the video is different from what it sees, or it follows the videos to do extra actions that are not required in the tasks. This suggests that future improvements can be made by improving the agent’s ability to adapt the selected trajectory to the testing task.
>
> Interestingly, we observe that very **few failures are caused by the mislabeled actions or incoherent trajectory objectives from the video processing stage**. We hypothesize that this is because our pipeline does not require the VLM to generate precise actions, but instead only needs high-level actions that are easier to annotate (e.g., click button A or type text B). Moreover, occasional mislabels often do not change the overall instructional content of the trajectory. For instance, Figure 9 in Appendix C shows an example where a mislabeled action (highlight instead of click at the last step) does not change the overall demonstration of the trajectory, and the agent still follows the trajectory to correctly complete the task.
>
> [2] Xie et al., OSWORLD: Benchmarking Multimodal Agents for Open-Ended Tasks in Real Computer Environments.

---

> > ### Author Response · Authors · 2025-12-13
> >
> > **3. Discuss robustness to noisy or misleading videos.**
> >
> > We conducted additional experiments and case studies. We find that our system is **more likely to be affected by videos that are similar to the task but contain misleading guidance**, and it is **relatively robust to completely irrelevant videos**.
> >
> > Firstly, in our added error analysis, we find a case where the video demonstrates a similar task (how to set the default language in Chrome), but the video’s UI version is different from the agent’s environment (the agent’s environment does not have the same button shown in the video). This discrepancy causes confusion and repeated clicks, which eventually lead to the failed task. Please refer to Figure 2 in Appendix B for the details.
> >
> > Next, we conducted an additional experiment with noisy videos. Specifically, instead of using videos after careful filtering, we take the videos that **do not pass** the first-step coarse selection of our video filtering. These videos are usually irrelevant to the task (e.g., for the task “Could you help me stretch this image to fill the entire page, keeping its proportion and centering the image?” on LibreOffice Impress, one video title is “How to insert picture into table in word”). We test our method’s performance when using these noisy videos while keeping everything else the same (same video processing and trajectory selection as our method). The following table shows the performance on a subset of 50 tasks in OSWorld.
> > |  | Success Rate |
> > |---|---|
> > | Ours | 52.0 |
> > | Ours (noisy videos) | 54.0 |
> >
> > As can be observed, our method with irrelevant videos actually completes one more correct task. These results demonstrate that our system is more robust to irrelevant videos, but it is more likely to be affected by the “adversarial videos” that look similar but contain misleading guidance.
> >
> > **4. Quantify video retrieval effectiveness.**
> >
> > The following table shows the fraction of initially retrieved videos that pass our coarse and content filtering.
> > |  | Pass rate over initially retrieved videos (%) |
> > |---|---|
> > | Coarse filtering | 21.4 |
> > | Content filtering | 12.9 |
> >
> > Additionally, we manually examine the final resulting videos after the two-step filtering on 10 random tasks. Among the 31 final videos, 28 of them are judged as relevant and helpful. These results collectively demonstrate the effectiveness of our video filtering procedure.
> >
> > **5. Add qualitative trajectory visualizations.**
> >
> > We have added **Appendix C** in our paper, which includes an example walking through our video processing pipeline. The added example shows how our pipeline identifies changing frames, labels actions, generates an objective that matches the task requirement, and filters invalid trajectories. Please refer to the updated PDF for more details.
> >
> > **6. Clarify scaling behavior and limits.**
> >
> > We didn’t test performance with more videos beyond the current number because we already retrieved the most possible videos per query (50 for YouTube API), and the current videos are what’s left after filtering. We will leave it for future work to further scale the number of videos. For example, one could generate more queries per task where each query focuses on different aspects of the task. This may retrieve more videos that demonstrate the relevant knowledge of the task. We have revised the paper to ensure that we don’t overclaim that increasing the number of videos will further improve the performance.
> >
> > However, with a larger demonstration pool, we believe our two-stage selection method has the potential to address associated issues, such as context length and selection noise, and further improve performance. Specifically, in the first stage selection, we only provide the agent with the textual objective of each trajectory, which is deeply compressed (around ten tokens per trajectory). This largely prevents the context explosion issue. Moreover, our second-stage selection performs a more detailed analysis using information such as the action list and the first frame of the trajectory, which largely addresses the selection noise issue.
> >
> > In summary, unless the demonstration pool is extremely large (hundreds of videos per task), we believe our method has the potential to further improve performance.
> >
> > **7. Improve reproducibility of experiments.**
> >
> > We will release all necessary data to reproduce our experiments, including the list of video URLs, the processed trajectories with objective, action list, and frames at each step. We will also release our code for video processing and agent inference.

---

> > > ### Author Response · Authors · 2025-12-13
> > >
> > > **8. Explicate assumptions on VLM capabilities.**
> > >
> > > As mentioned in our response to **Q2**, we observe that very few failures are caused by the mislabeled actions or incoherent trajectory objectives from the video processing stage, since our pipeline only needs high-level actions that are easier to annotate. This reduces dependence on extremely strong VLMs. In our experiments, we use Qwen2.5-VL-32B for processing. In our preliminary experiments with a smaller model (Qwen2.5-VL-7B), we observed substantially noisier and less reliable action labeling. Therefore, in practice, we expect the VLM to be comparable to or stronger than Qwen2.5-VL-32B for robust operation.
> > >
> > > **Broader Impact Concerns**
> > >
> > > Thanks for the suggestions. We have updated our paper with section **6. Broader Impact** to address these concerns.

---

### Decision · Action_Editor_ho2Q · 2025-12-22

**Recommendation:** Accept with minor revision

**Additional Comments:**

All reviewers acknowledged the clear motivation addressing how agents lack domain-specific procedural knowledge, which humans typically acquire by watching tutorial videos. The pipeline is well-structured with three modular stages (retrieval, processing, application) that integrate easily with existing agent frameworks. The authors have responded comprehensively to reviewer concerns through detailed computational analysis, thorough failure mode investigation, and extensive new experimental results during the rebuttal.

I think the concerns of the cost for practical deployment is reasonable. So I would like to ask the authors to make the following changes for the final publication:
1. The authors should add a dedicated section or subsection explicitly addressing computational practicality and deployment scenarios, including a more detailed discussion about the cost-benefit analysis.
2. The authors should carefully revise the claims made in this paper to avoid the over-claiming, given that the potential cost overhead brought by this method.
3. Adding a limitation section to discuss these issues, which is important for readers to understand the methods' boundary.

Overall, this paper is a good paper and I will recommend accept with minor revision.

**Audience:**

Yes

**Audience Explanation:**

Computer use is an important topic, definitely this work is relevant to the TMLR community.

**Claims And Evidence:**

Yes

**Claims Explanation:**

The submission includes sufficient experiment results to support their claims.